# Regain flood adaptation in rice through a 14-3-3 protein OsGF14h

Jian Sun [1,2] ✉, Guangchen Zhang[1,2], Zhibo Cui[1,2], Ximan Kong[1,2], Xiaoyu Yu[1], Rui Gui[1], Yuqing Han[1], Zhuan Li[1], Hong Lang[1], Yuchen Hua[1], Xuemin Zhang[1], Quan Xu[1], Liang Tang[1], Zhengjin Xu[1], Dianrong Ma[1] & Wenfu Chen [1] ✉

Contemporary climatic stress seriously affects rice production. Unfortunately, long-term domestication and improvement modified the phytohormones network to achieve the production needs of cultivated rice, thus leading to a decrease in adaptation. Here, we identify a 14-3-3 protein-coding gene *OsGF14h* in weedy rice that confers anaerobic germination and anaerobic seedling development tolerance. *OsGF14h* acts as a signal switch to balance ABA signaling and GA biosynthesis by interacting with the transcription factors OsHOX3 and OsVP1, thereby boosting the seeding rate from 13.5% to 60.5% for anaerobic sensitive variety under flooded direct-seeded conditions. Meanwhile, *OsGF14h* co-inheritance with the *Rc* (red pericarp gene) promotes divergence between temperate *japonica* cultivated rice and temperate *japonica* weedy rice through artificial and natural selection. Our study retrieves a superior allele that has been lost during modern *japonica* rice improvement and provides a fine-tuning tool to improve flood adaptation for elite rice varieties.

As current climatic stress limits yield increasingly, more consistent crop production must be achieved[1,2]. A total inundation area of 2.23 million square kilometers, with 255 to 290 million people directly affected by floods in 913 large flood events worldwide from 2000 to 2018[3], and agricultural production has also been severely damaged. Direct seeding is a vital cultivation method for rice that is increasingly employed in both rainfed and irrigated fields due to its advantages of reduced labor, energy, water use, production costs, and mechanization[4–6]. Heavy rainfall and floods seriously affect crop production, especially germination and seedling development of direct-seeded rice (DSR), which requires that varieties exhibit uniform and rapid germination to establish seedlings in anaerobic conditions caused by flooding. However, the intensive domestication and improvement have led to a decrease in flood adaptation and poor capacity for anaerobic germination (AG)/anaerobic seedling development (ASD)[7] to achieve the production needs of cultivated rice[1,8,9]. Capturing natural genetic variation from germplasm resources and deciphering the mechanism(s) regulating AG/ASD is an opportunity to improve the status quo.

The *SUB1A*, which encodes an ethylene-responsive factor-type transcription factor (ERF), confers submergence tolerance to rice by limiting gibberellin (GA) response, thereby inhibiting plant elongation and sustaining seedling vigor for up to two weeks under submerged conditions[10,11]. By contrast, deepwater rice, with the *SNORKEL 1* and *SNORKEL 2* genes encode *SUB1A*-like transcription factors, can invest photo-assimilate into the extension of submerged stem internodes by GA biosynthesis[12]. In addition, under continuous flooding, the transcript accumulation of *SD1*, known as the 'green revolution gene', induced by ethylene signaling, activates GA synthesis and internode growth of stems, allowing deepwater rice to survive[13]. *OsGA20ox1*, a paralog of *SD1*, also has positive functions on seedling vigor in the initial growth of rice[14].

Seed germination requires coordination of embryonic development and environmental conditions, in which the balance of two competing hormone signaling pathways, abscisic acid and gibberellin, plays a decisive role[15]. Although seedling-stage submergence tolerance and avoidance mechanisms ensure rice survival, the mechanism(s) for ensuring uniform and rapid seed germination under flooded-

[1]Rice Research Institute, Shenyang Agricultural University, Shenyang 110866, China. [2]These authors contributed equally: Jian Sun, Guangchen Zhang, Zhibo Cui, Ximan Kong. ✉e-mail: sunjian811119@syau.edu.cn; wfchen@syau.edu.cn

anaerobic conditions remains explored. A trehalose-6-phosphate phosphatase gene, *OsTPP7* can enhance starch mobilization to drive growth kinetics of the germinating embryo and elongating coleoptile[5]. Another AG gene, *OsCBL10*, acts as a negative regulator of the CIPK15-dependent low-oxygen signaling pathway to enhance total α-amylase activity[16]. Screening germplasms with strong AG/ASD tolerance and isolating more functional genes are required for DSR development. They will contribute to the understanding of the molecular mechanism(s) underlying the flood adaptation in rice.

Weedy rice (*Oryza sativa* f. *spontanea*) competes aggressively with cultivated rice as an invasive paddy weed. It reduces harvests, which is also reputed to be a 'hidden gold mine' in global paddy fields due to the substantial stress resistance and the nutritional quality different from cultivated rice[17–21]. In addition, the origin and evolution of weedy rice have been extensively studied based on population genomic investigations in recent years[18,22,23]. Here, we identify a 14-3-3 protein-coding gene *OsGF14h* in weedy rice, which acts as a signal switch to inhibit ABA signaling through interacting with the transcription factors OsHOX3 and OsVP1 and enhance GA synthesis, thereby improving AG and ASD. *OsGF14h* can improve flood adaptation for elite rice varieties as a fine-tuning tool. We further reveal the critical role of *OsGF14h* in rice genetic improvement and ecotype differentiation, whereby providing an empirical example of how the genetic network of phytohormones responds to different selection pressures.

## Results

### *OsGF14h* isolated from weedy rice is associated with flood adaptation

After years of investigation, we found that weedy rice spontaneously appeared in paddy fields after cultivated rice was transplanted in Northeast China. Therefore, weedy rice can be regarded as a natural DSR due to its AG tolerance and successful seedling establishment under anaerobic conditions. To isolate essential AG tolerance genes in weedy rice, we first used recombinant inbred lines (RILs) population with 168 lines (generation $F_8$) that derived from AG tolerant weedy rice WR04-6 and AG susceptible cultivated rice Qishanzhan to perform QTL mapping analysis. This population has been effectively used to study the genetic basis of agronomic traits in weedy rice (Supplementary Data 1)[22]. Based on the inclusive composite interval mapping (ICIM) method with a LOD threshold of 2.5, we detected two QTLs. The main QTL was mapped in the physical range of chromosome 11 from 23,790,182 to 23,859,119 in the *japonica* reference genome (Os-Nipponbare-Reference-IRGSP-1.0) with a LOD value of 5.65 and phenotypic variance explained (PVE) of 15.85 (Fig. 1a).

We also constructed a genome-wide association study (GWAS) panel including 36 temperate *japonica* weedy rice (*GJ-tmp-weedy*), 57 temperate *japonica* landrace (*GJ-tmp-land*), and 97 temperate *japonica* cultivar (*GJ-tmp-cul*) to detect AG associated QTNs (quantitative trait nucleotides) (Supplementary Data 2). The mean germination rate of the accessions in the GWAS panel was in the order of weedy rice (40.05%) > landrace (31.48%) > cultivated rice (26.08%) after 72 h of anaerobic incubation (Fig. 1c). The GWAS based on the linear mixed model with the first two principal components from principal components analysis (PCA) as covariates (Fig. 1d) showed that the strongest associated signals were also detected on chromosome 11 with the ideal distribution of observed *P* value based on QQ plot analysis (Fig. 1e); the physical position of QTN peaks occurs at 23,556,546 bp (Fig. 1b) with the $-\log_{10}(P)$ value of 8.279. Considering the mapping results of QTL and GWAS, we speculate that genes in this genomic region contribute to the AG tolerance of weedy rice. Then we plotted the linkage disequilibrium (LD) block around the target QTN peak (Fig. 1f) and preselected causal haplotypes of candidate genes that were annotated by MSU Rice Genome Annotation Project Release 7 within the LD block. (Supplementary Fig. 1a–g). Finally, the candidate genes were narrowed down to seven within a 584 kb genomic region,

*LOC_Os11g39370, LOC_Os11g39450, LOC_Os11g39500, LOC_Os11g39520, LOC_Os11g39530, LOC_Os11g39780*, and *LOC_Os11g39540*.

*LOC_Os11g39540* and *LOC_Os11g39370* showed the highest expression level among the candidate genes under anaerobic stress (Supplementary Fig. 1h), whereas *LOC_Os11g39370* is mainly expressed in leaves and *LOC_Os11g39540* is mainly expressed in seed based on two Spatio-temporal expression databases (Supplementary Fig. 2). Thus *LOC_Os11g39540* is considered to be the final candidate gene by considering the gene expression model and annotation (Supplementary Data 3). *LOC_Os11g39540* gene encodes a 14-3-3 protein and is named *OsGF14h*, which is the eighth gene in its family based on gene annotation (http://rice.uga.edu/). Comparing cDNA sequences between an AG tolerant weedy rice (WR04-6) and a super-high-yield but AG-sensitive *japonica* variety (Shennong9816, SN9816) showed that the coding region of *OsGF14h* in SN9816 was consistent with the Nipponbare (IRGSP-1.0). SN9816 has a total of six polymorphic sites in the coding region compared with WR04-6: one deletion of four bases at the fourth exon of SN9816 causes frameshift mutations and stops amino acid translation; two non-synonymous substitution SNPs appear before the 4 bp deletion; one non-synonymous substitution SNP appears after the 4 bp deletion; and two synonymous substitution SNPs, which results in an incomplete 14-3-3 protein (Fig. 1g). Therefore, we predicted that this natural variation of *OsGF14h* may have a genetic effect on AG tolerance. The best linear unbiased prediction (BLUP) value of each individual in the GWAS panel to show the corrected allelic effects with the correlation coefficient with the phenotype is 0.603 (Fig. 1c), and the difference in BLUP value between the two groups classified based on 4 bp deletion is more obvious (Fig. 1h).

### *OsGF14h* promotes AG tolerance and ASD in rice

Based on annotations, *OsGF14h* may act as a chaperone that participates in protein modification and signal regulation (Supplementary Data 3). In WR04-6, the qPCR analysis indicated that *OsGF14h* transcripts were detected in all tissues and were highly expressed in the developing seeds and endosperm (Fig. 2a). The OsGF14h[WR04-6]::GFP fusion protein is localized in both the nucleus and cytoplasm (Fig. 2b).

To further study the role of *OsGF14h* on AG tolerance and ASD, we introduced the coding sequence (CDS) from *OsGF14h*[WR04-6] into SN9816 (WT[SN9816]) under the control of the CaMV 35 S promoter, whereby two homozygous overexpression lines Ox*OsGF14h*[WR04-6]-8 and Ox*OsGF14h*[WR04-6]-25 were used in the present study. Meanwhile, another two overexpression lines of WT[SN9816] with its own CDS overexpressed, Ox*OsGF14h*[SN9816]-5 and Ox*OsGF14h*[SN9816]-11, were created. We also used a CRISPR/Cas9 system to specifically disrupt the *OsGF14h* gene in WR04-6 (WT[WR04-6]) (Supplementary Fig. 12a), whereby two mutants (*gf14h-13* and *gf14h-15*) with other overexpression lines were used to the comparison of genetic effects in the present study. Compared with WR04-6, the two mutants (*gf14h-13* and *gf14h-15*) showed reduced coleoptile elongation and seedling size under anaerobic conditions, whereas the overexpression lines (Ox*OsGF14h*[WR04-6]-8 and Ox*OsGF14h*[WR04-6]-25) showed significantly enhanced AG ability and seedling vigor as compared with WT[SN9816] (Fig. 2c–e), however, the other two overexpression lines (Ox*OsGF14h*[SN9816]-5 and Ox*OsGF14h*[SN9816]-11) showed no significant difference from of WT[SN9816] (Fig. 2d). These results indicated that *OsGF14h* was the target gene promoting AG tolerance and ASD in rice. Additionally, we found that overexpression of *OsGF14h* boosted the seeding rate from 13.5 to 60.5% under the flooded direct-seeded condition in outdoor anaerobic facilities (Fig. 2f, g).

### *OsGF14h* interacts with OsHOX3 and OsVP1 to inhibit the ABA signaling

To better understand the mechanism(s) of how *OsGF14h* improve AG, we performed yeast two-hybrid (Y2H) screening with pGBKT7-OsGF14h as bait. In total, 31 candidate proteins were identified to

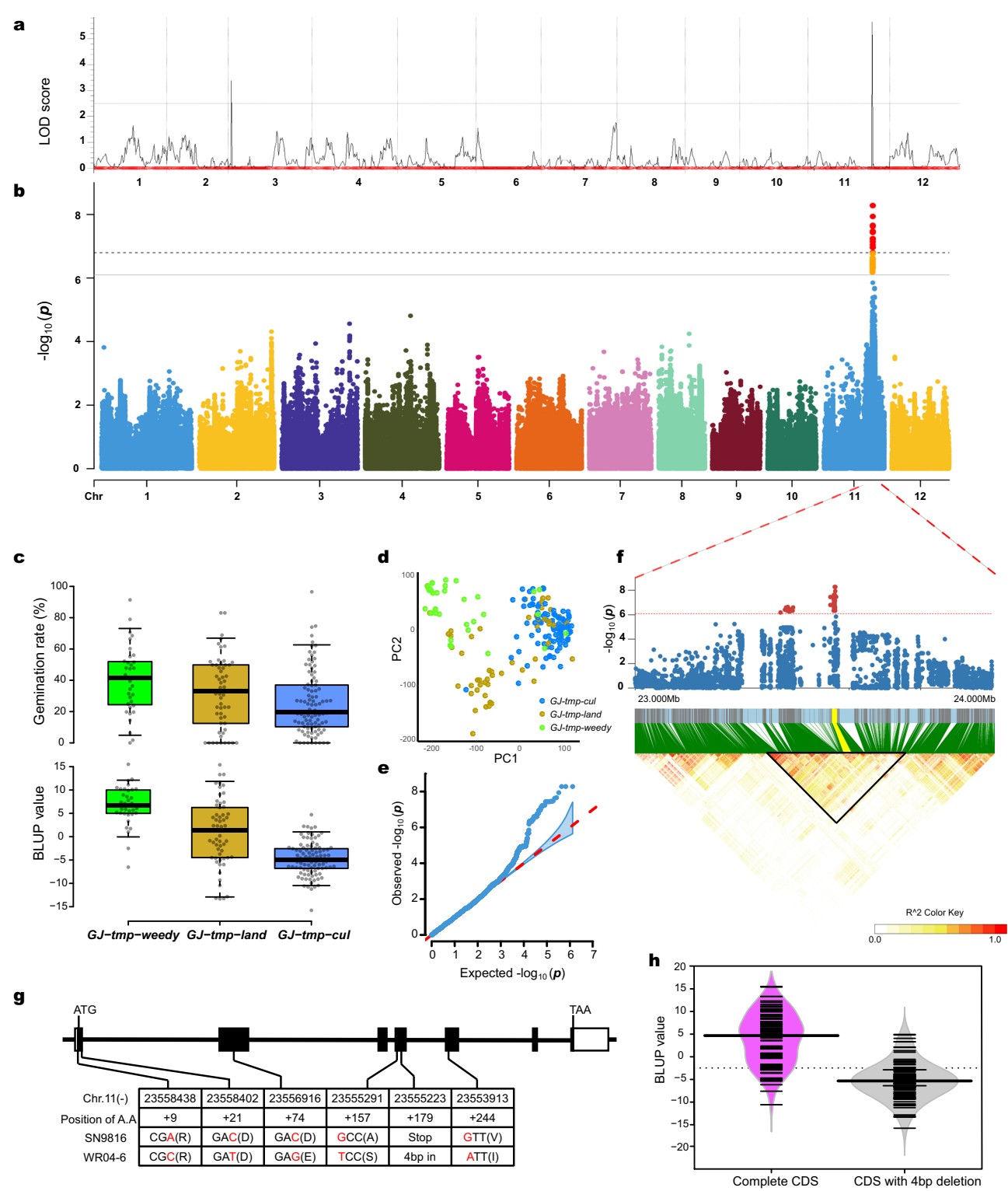

interact with OsGF14h^(WR04-6), including an HD-Zip family TF OsHOX3 (*LOC_Os01g45570*), which was reported as a transcription-repressor related to GA regulation[24] and an ABA crucial transcription activator OsVP1 (*LOC_Os01g68370*)[25]. These two interactions can be verified by transient expression in the luciferase (LUC) complementation imaging assays, and we also confirmed that both OsGF14h^(WR04-6) and OsGF14h^(SN9816) can interact with OsHOX3 and OsVP1 in vivo and in vitro by using Y2H and pull-down assays respectively (Fig. 3a–c). Based on the prediction by AlphaFold2[26], OsGF14h^(WR04-6) encodes a fully functional 14-3-3 protein that, with nine alpha-helix structures, could form a

clamp to bind the N-terminal loop of OsHOX3. For the OsGF14^(SN9816), the frameshift mutation caused by the 4 bp deletion in the coding region led to a loss of one helix in the C-terminus, which may weaken the strength of its interaction with OsHOX3 (Supplementary Fig. 3a, b). The co-immunoprecipitation (co-IP) and Y2H assays further proved the strength of the two interactions between OsGF14h^(WR04-6) and OsHOX3 and between OsGF14h^(WR04-6) and OsVP1 was greater than that of OsGF14h^(SN9816) in vivo (Fig. 3a, d).

We also regenerated an overexpression construct (*35 S: OsVP1-GFP*) and transformed it into WT^(WR04-6), then two overexpression lines

**Fig. 1 | *OsGF14h* isolated from weedy rice is associated with anaerobic germination (AG) based on QTL mapping and GWAS. a** QTL mapping results in a LOD value threshold higher than 2.5. **b** Manhattan plot of GWAS for AG with a threshold of $8.12 \times 10^{-7}$ (0.05 significance level) and $1.62 \times 10^{-7}$ (0.01 significance level). **c** Box plots for germination rate after 72-h anaerobic treatment (upper) and BLUP value among *GJ-tmp-weedy*, *GJ-tmp-land*, and *GJ-tmp-cul* from the GWAS panel (lower). Dots represent the accessions, $n = 190$ biologically independent samples. Center lines show the medians, box limits indicate the 25th and 75th percentiles, whiskers extend to the 5th and 95th percentiles as determined by R software. **d** PCA plot for the GWAS panel and dots represent the accessions, $n = 190$ biologically independent samples. **e** Quantile-quantile plots (QQ plots) of GWAS and plots corresponding to the GWAS Manhattan plot. **f** Linkage disequilibrium plot for SNPs

within 994.84 kb genomic region around QTN peak detected by GWAS. Blue bars are SNP positions with *OsGF14h* highlighted by yellow. LD block is marked by a black triangle. The color key (white to red) represents linkage disequilibrium values ($R^2$). **g** Gene structure and haplotypes between WR04-6 and Shennong9816. **h** Bean plot indicates BLUP value between the two groups of GWAS panel that are classified by the 4 bp deletion. In the group of the pink bean, all accessions' CDS does not have 4 bp deletion in *OsGF14h* CDS ($n = 89$ biologically independent samples), and accessions in the gray group have 4 bp deletion in *OsGF14h* CDS ($n = 101$ biologically independent samples). Short black lines within polygons represent individual BLUP values, long black lines show the medians, and polygons represent the estimated density of the data. BLUP values for **c**, **h** are provided in Supplementary Data 2.

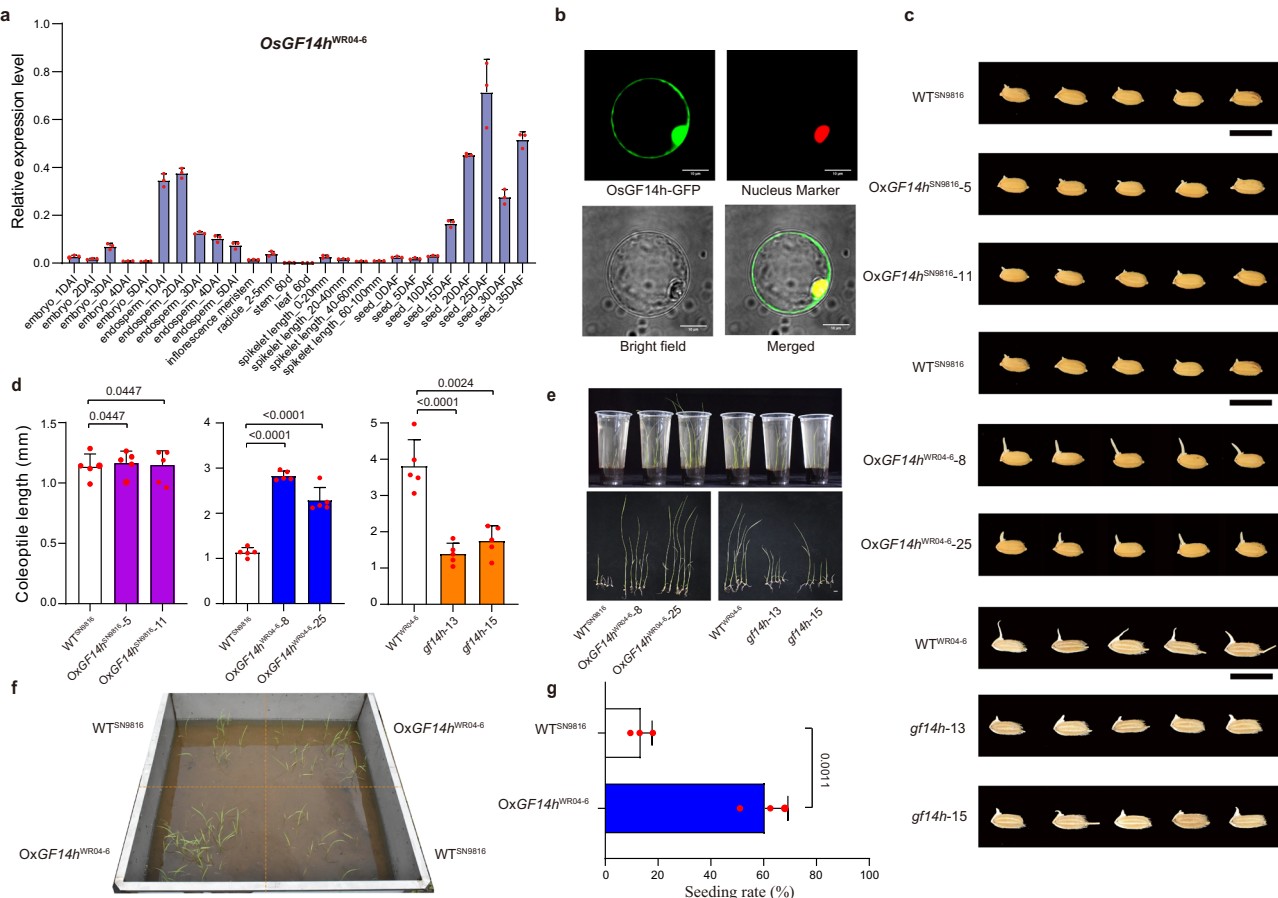

**Fig. 2 | Genetic evidence and effects of *OsGF14h* in response to the anaerobic stress. a** qRT-PCR-based transcript abundance analysis of *OsGF14h* in organs and tissues of WR04-6, including the germinating embryo and endosperm, developing seeds, radicle, stem, leaf, spikelet, and inflorescence meristem. DAI days after imbibition, DAF days after fertilization. Data were presented as mean ± SD, $n = 3$ biologically independent samples. **b** Subcellular localization of OsGF14h^WR04-6-GFP fusion protein in rice protoplasts. Scale bars = 10 μm. A representative experiment from three independent experiments is shown. **c** Performance of anaerobic germination between WT^SN9816 and its overexpression lines (Ox*OsGF14h*^SN9816-5 and Ox*OsGF14h*^SN9816-11), between WT^SN9816 and its overexpression lines (Ox*OsGF14h*^WR04-6-8 and Ox*OsGF14h*^WR04-6-25), and between WT^WR04-6 and its CRISPR/Cas9 knockout lines (*gf14h*-13 and *gf14h*-15) after 4 days of anaerobic treatment. Scale bar, 1 cm. **d** The comparisons of coleoptile lengths between WT^SN9816 and its overexpression lines Ox*OsGF14h*^SN9816, between WT^SN9816

and Ox*OsGF14h*^WR04-6, and between WT^WR04-6 and its knockout lines after 4 days of anaerobic germination, respectively. Data were presented as means ± SD. $n = 5$ independent experiments. *P* values are indicated by an unpaired two-tailed Student's *t*-test. **e** Performance of anaerobic seedling development between WT^SN9816 and overexpression lines Ox*OsGF14h*^WR04-6, between WT^WR04-6 and knockout lines under the flooded direct-seeded condition on the 21st day. Scale bar, 1 cm. **f** Performance of flooding adaptation of WT^SN9816 and its overexpression line Ox*OsGF14h*^WR04-6 in flooded tanks under flooded direct-seeded condition (a water layer of 10 cm was maintained); the photo was taken on the 21st day after flooding. **g** The seeding rate of WT^SN9816 and Ox*OsGF14h*^WR04-6 seedlings. Data were presented as means ± SD. $n = 3$ independent measurements in three flooded tanks. *P* values are indicated by an unpaired two-tailed Student's *t*-test. Source data are provided as a Source Data file.

(Ox*VP1*-3 and Ox*VP1*-5) that were used together with the existing loss-of-function mutant of *OsHOX3* (*hox3*) to verify their physiological functions in AG. The germination and coleoptile elongation of Ox*VP1* lines and *hox3* were found to be blocked compared with their wild-type WT^WR04-6 and WT^kitaake respectively (Fig. 3e, f). Previous studies reported

that OsVP1 could not bind ABA-responsive elements (ABREs) independently, instead of interacting with TRAB1 (a bZIP-TF that can interact with both OsVP1 and ABREs) to inhibit ABA response[25]. Significantly, 14-3-3 protein mediates the trans-regulation of the ABA-responsive factor OsEM1 by OsVP1[27]. In the present study, the

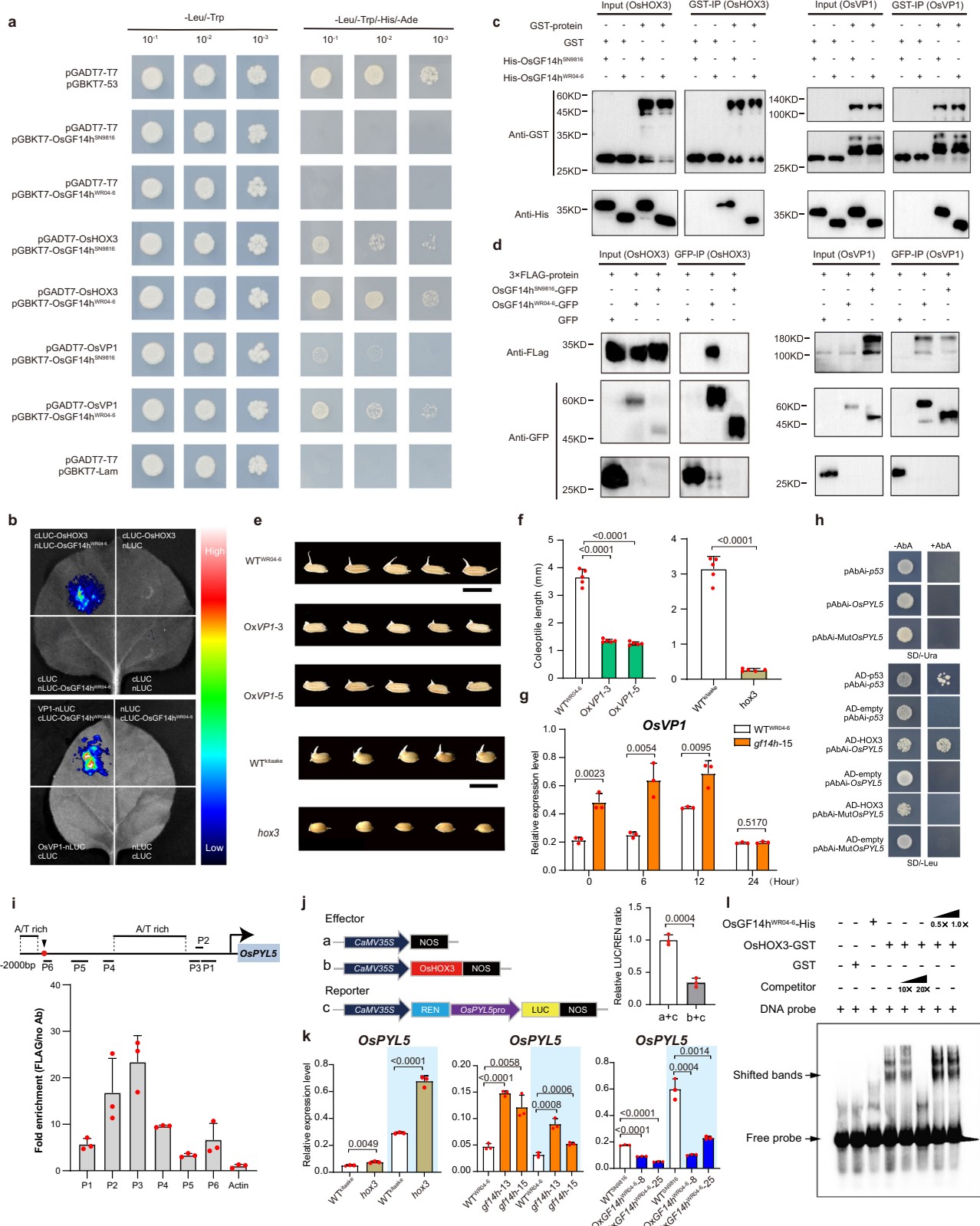

expression level of *OsVP1* in the *OsGF14h* knockout line (*gf14h-15*) was significantly greater than that in WT^WR04-6 during the initial stage and then significantly reduced at the 24th hour after it was flooded (Fig. 3g). Meanwhile, the *TRAB1* expression also appears to be negatively regulated by the *OsGF14h*^WR04-6 (Fig. 4a). The above evidence implies that a stronger interaction between OsGF14h^WR04-6 and OsVP1 can inactivate the ABA response.

The favored recognition site of OsHOX3 was shown to be composed of two 5-bp half-sites that overlap at a central position, 'CAAT(G/C)ATTG'[28]. Based on the prediction of cis-acting elements for the target genes of the OsHOX3 protein, we found that *OsPYL5/RCAR5*, an ABA receptor in the ABA signaling unit[29], was one of the target genes of OsHOX3. Based on yeast one-hybrid (Y1H) assays, we found that OsHOX3 can specifically bind the HD-ZIP3 element in the *OsPYL5*

**Fig. 3 | OsGF14h interacts with OsHOX3 and OsVP1 to modulate the ABA-responsive pathway. a** Y2H assay. The interaction between (OsGF14h$^{WR04-6}$ and OsGF14h$^{SN9816}$) and (OsHOX3 and OsVP1). Strains carrying two indicated constructs were grown on a synthetic medium. **b** Luciferase complements imaging assays. *Agrobacterium* carrying different plasmids, as indicated, were co-expressed in *Nicotiana benthamiana*, the colors representing interaction strength. **c** Pull-down assay. GST-OsHOX3 and GST-OsVP1 were used as baits and the pull down of His-OsGF14h$^{WR04-6}$ and His-OsGF14h$^{SN9816}$ were detected by the anti-His antibody. **d** Co-IP assay. Input means protein number before the experiment. GFP-IP means the protein number detected after co-IP buffer washing. **e, f** Performance (**e**) and coleoptile length (**f**) of WT$^{WR04-6}$ and two overexpression lines, and WT$^{kiaake}$ and mutant under anaerobic stress. Scale bar = 1 cm. *n* = 5 biologically independent experiments. **g** Relative expression of *OsVP1* in WT$^{WR04-6}$ and knockout line at different hours under anaerobic stress. *n* = 3 biologically independent samples. **h** Y1H assay showing the interaction between OsHOX3 and the promoter of *OsPYL5*. -AbA, 0 ng/mL; +AbA, 200 ng/mL. **i** ChIP-qPCR assays. Enrichment of OsHOX3 in *OsPYL5* promoter. The upper schematic indicates the locations of the DNA fragments used for ChIP-qPCR. The red square indicates the element. A/T rich indicates a high A/T base region. Error bars indicate the means ± SD for independent experiments. **j** Dual-LUC assays. OsHOX3 binding *OsPYL5* promoter. The schematic indicates effector and reporter constructs. **k** Expression of *OsPYL5* at the 12th hour of WT$^{kiaake}$ and *hox3* mutant, and at the 36th hour of WT$^{WR04-6}$, knockout line and expression lines under anaerobic (blue background) and aerobic conditions. *n* = 3 biologically independent samples. **l** EMSA assay. The competitor was unlabeled and the concentrations were 10× and 20× of the biotin-labeled probe. 0.5× and 1× OsGF14h$^{WR04-6}$ protein amount of OsHOX3 was added. In **a, b, c, d, h, i, j, l**, representative experiments from three independent experiments are shown. In **f, g, j, k** data were presented as means ± SD, *P* values are indicated by unpaired two-tailed Student's *t*-test. Source data are provided as a Source Data file.

promoter region (Fig. 3h). Then, the ChIP-qPCR analysis revealed three enrichments of OsHOX3 in the *OsPYL5* promoter segments at P2, P3, and P4 (Fig. 3i). The dual-luciferase (LUC) assay further showed that co-expression of *35 S: OsHOX3* and *OsPYLSpro: LUC* significantly reduced the LUC/REN ratio compared to the control (Fig. 3j). The *OsPYL5* transcription level was significantly increased in the mutant *hox3* compared to its wild-type (WT$^{kitaake}$) (Fig. 3k). Meanwhile its expression also correspondingly exhibited an increase in knockout lines (*gf14h-13* and *gf14h-15*) and a decrease in overexpression lines (Ox*OsGF14h*$^{WR04-6}$-8 and Ox*OsGF14h*$^{WR04-6}$-25) compared to their wild types (Fig. 3k). To further reveal whether OsGF14h is directly involved in OsHOX3 moduling OsPYL5, we performed electrophoretic mobility shift assays (EMSA). The result showed that the binding capacity of OsHOX3 to the *OsPYL5* promoter was evidently enhanced when adding OsGF14h$^{WR04-6}$ protein, indicating that OsGF14h$^{WR04-6}$ contributes to the binding (Fig. 3l).

To further reveal the response of the ABA signaling pathway under the regulation of *OsGF14h*, in addition to *OsTRAB1*, we also detected the expression levels of two bZIP-TF genes that are related to ABA-regulation (*OsOREB* and *OsbZIP72*) and two typical ABA-responsive genes (*OsEM1* and *OsRab16A*), between knockout lines (*gf14h-13* and *gf14h-15*) and WT$^{WR04-6}$, and between overexpression lines (Ox*OsGF14h*$^{WR04-6}$-8 and Ox*OsGF14h*$^{WR04-6}$-25) and WT$^{SN9816}$ under anaerobic condition. The results showed that most of these genes were inducted by flooding at the 24 and 48 h (Supplementary Data 4) and the expressions were suppressed significantly by *OsGF14h* under anaerobic conditions at the 12 h (Fig. 4a). An assay of exogenous ABA with gradient concentrations demonstrated that the sensitivity of the knockout lines (*gf14h-13* and *gf14h-15*) was higher than WT$^{WR04-6}$ (Fig. 4b). The relative germination rate (germination rate under the treatment of exogenous 1 μM ABA divided by that of under normal conditions) further verified that the ABA sensitivity of two overexpression lines (Ox*OsGF14h*$^{WR04-6}$-8 and Ox*OsGF14h*$^{WR04-6}$-25) was higher compared with WT$^{SN9816}$. In contrast, the other two overexpression lines (Ox*OsGF14h*$^{SN9816}$-5 and Ox*OsGF14h*$^{SN9816}$-11) also showed a consistent level of ABA sensitivity with the WT$^{SN9816}$ (Fig. 4c). On the other hand, the content of endogenous ABA was not significantly different between knockout lines *gf14h-15* and WT$^{WR04-6}$ and between two kinds of overexpression lines Ox*OsGF14h*$^{SN9816}$-5 and Ox*OsGF14h*$^{WR04-6}$-8 (Supplementary Fig. 4). The above experimental evidence reveals an ABA signaling pathway in which *OsGF14h* acts as a signal switch, and OsGF14h interacts with OsHOX3 and OsVP1 to inhibit ABA response and improving flood adaptation.

### OsGF14h enhances GA biosynthesis
RNA-seq for Ox*OsGF14h*$^{WR04-6}$-8 and Ox*OsGF14h*$^{SN9816}$-5 under both anaerobic and aerobic conditions were conducted for evaluated genome-wide effects of *OsGF14h*. The results showed that the number of differential expressed genes (DEGs) (Ox*OsGF14h*$^{WR04-6}$-8/

Ox*OsGF14h*$^{SN9816}$-5 > 2 and FDR <0.01) between Ox*OsGF14h*$^{WR04-6}$-8 and Ox*OsGF14h*$^{SN9816}$-5 was much higher under anaerobic conditions (4183 DEGs) than under aerobic conditions (84 DEGs) (Supplementary Fig. 5a). Genes involved in many metabolic pathways that related to catalytic activity, binding, and transporter activity among these 4183 DEGs were enriched based on GO and KEGG analysis (Supplementary Fig. 5b, c). Genes involved in gibberellin biosynthesis are crucial for rice development, especially *OsGA20ox1* play an important role in the initial growth stage[14]. We then focused on 40 GA biosynthesis genes of the anaerobic group and found that all six DEGs, including OsGA20ox1, among the 11 detectable positively regulated genes were upregulated, two DEGs among the five detectable positively regulated genes were upregulated, and the other nine genes were not differentially expressed (Fig. 4d). On the other hand, among 74 detectable genes of 94 ABA-responsive genes, 13 DEGs were downregulated, four DEGs were upregulated in Ox*OsGF14h*$^{WR04-6}$-8, and 57 genes were not differentially expressed (Supplementary Fig. 6). The expression of *OsGA20ox1* in knockout lines (*gf14h*-13 and *gf14h*-15), decreased significantly at 12 h under anaerobic conditions compared with that in WT$^{WR04-6}$. On the contrary, *OsGA20ox1* in overexpression lines (Ox*OsGF14h*$^{WR04-6}$-8 and Ox*OsGF14h*$^{WR04-6}$-25) increased significantly compared with WT$^{WR04-6}$ (Fig. 4e). We thus monitored the dynamic changes of four kinds of endogenous GAs (GA9, GA20, GA24, and GA53) that related to the *OsGA20ox1* pathway during the germination of WT$^{WR04-6}$ and *gf14h*-15 under anaerobic conditions. Compared with WT seeds, GAs' contents in *gf14h*-15 seeds showed a decreasing trend from 0 to 24 h after anaerobic incubation (Fig. 4f). This implies that *OsGF14h*$^{WR04-6}$ may promote the germination of weedy rice under anaerobic stress by enhancing GAs synthesis. To further verify the genetic function of *OsGA20ox1* for anaerobic germination, we generated CRISPR-Cas9-based knockout lines of *OsGA20ox1* (*ga20ox1*-1 and *ga20ox1*-3) in WT$^{WR04-6}$ (Supplementary Fig. 12b) and found that the knockout lines showed significantly reduced coleoptile length compared with WT$^{WR04-6}$ in anaerobic conditions (Fig. 4g). *OsHOX3* has been reported as a positive regulator in GAs biosynthetic pathway in the seedling stage[16], which has also been partially validated by qPCR during AG in this study. The results showed that the positive regulation of *OsHOX3* on the GA biosynthesis was mainly reflected in the degradation pathway, in addition to *OsGA20ox1* and *KOS4* in the biosynthesis pathway (Supplementary Fig. 7), *OsGF14h* can enhance GAs biosynthesis via but not limited to OsHOX3 thereby acts as a role in the balance of ABA and GA during anaerobic germination.

### Haplotype distribution and evolution of *OsGF14h*
To further reveal the haplotype distribution and evolution of *OsGF14h* in different rice ecotypes, we analyzed the *OsGF14h* haplotype frequency and network using sequence data of 1,596 accessions of *Oryza rufipogon* (*Or-IIIa*) and *O. sativa* from nine ecotypes with the *O.bathii* as

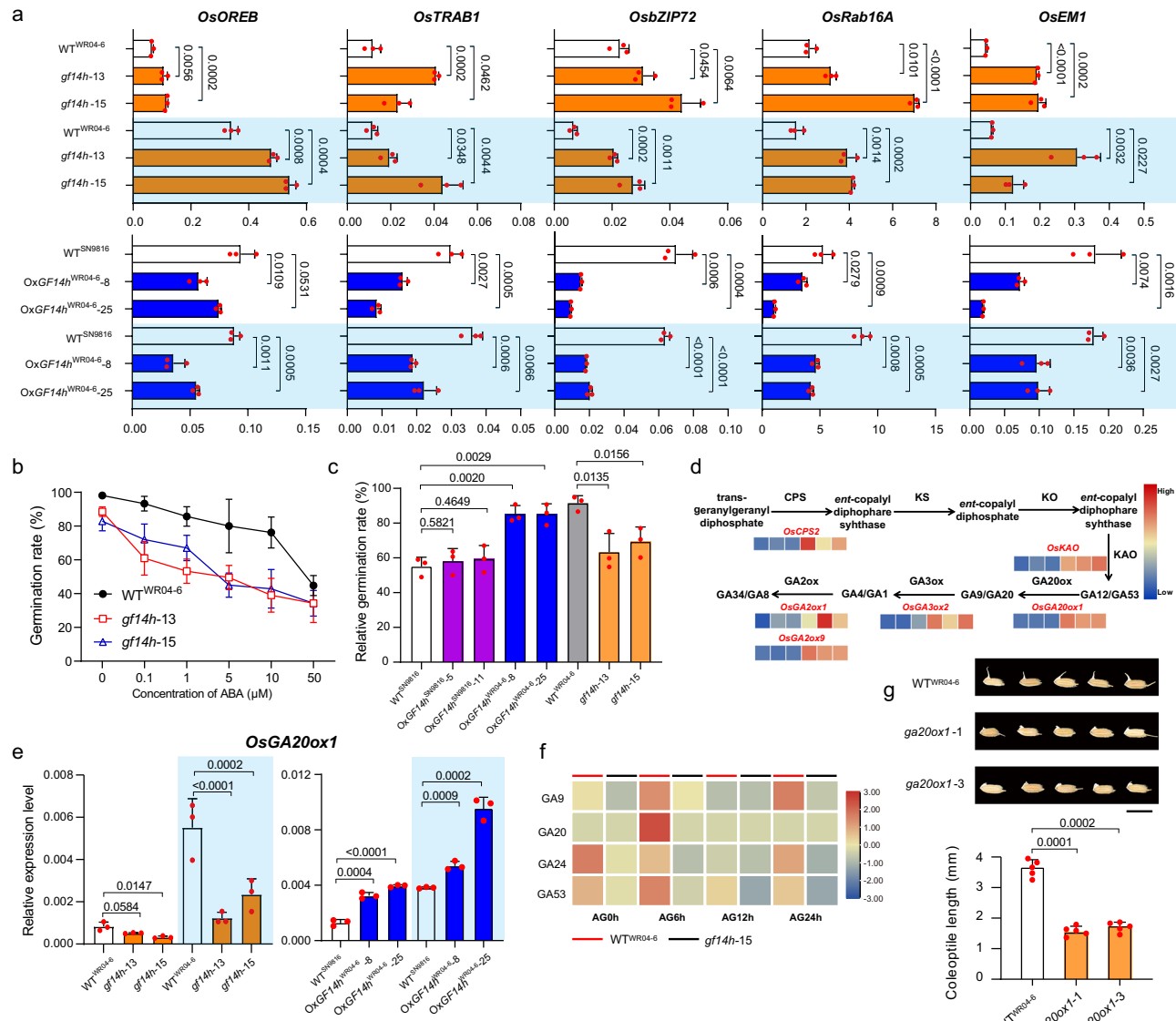

**Fig. 4 | *OsGF14h*^WR04-6 inhibits ABA response and enhances GA biosynthesis.**
**a** The relative expression of ABA regulated and responsive genes between WT^WR04-6 and *OsGF14h* knockout lines, between WT^SN9816 and *OsGF14h* overexpression lines under anaerobic (blue background) and aerobic conditions at the 12 h. **b** Assay of exogenous ABA with gradient concentrations to reflect the ABA sensitivity for *OsGF14h* knockout lines and WT^WR04-6. Error bars indicate the means ± SD of three independent experiments. **c** Comparison of relative germination rate between WT^SN9816 and two types of overexpression lines (function and partial loss-of-function) and between WT^WR04-6 and *OsGF14h* knockout lines under 72 h of anaerobic incubation with exogenous 1 µM ABA treatment. In **b**, **c** seeds with the embryo showing white without the emergence of coleoptile are still considered as germination. **d** The expression model of six DEGs on the GA biosynthesis pathway is presented based on RNA sequencing. The left three and right three rectangles under the gene names (highlighted in red) represent the FPKM value of the three

biological replicates of Ox*OsGF14h*^WR04-6 and Ox*OsGF14h*^SN9816, respectively, in response to anaerobic stress. The color scale indicates low (blue) to high (red) FPKM value. **e** The *OsGA20ox1* expression level between WT^WR04-6 and *OsGF14h* knockout lines and between WT^SN9816 and functional *OsGF14h* overexpression lines at the 12th hour under anaerobic (blue background) and aerobic conditions. **f** Heatmap of three kinds of endogenous GA content in WT^WR04-6 and *OsGF14h* knockout line *gf14h*-15 during anaerobic germination. The color key (blue to red) represents the GAs content from low to high value. **g** Performance and coleoptile lengths of WT^WR04-6 and *OsGA20ox1* knockout lines on the 4th day of anaerobic germination. Scale bar = 1 cm. In **a**, **c**, **e**, **g**, data were presented as means ± SD, P values are indicated by an unpaired two-tailed Student's *t*-test. In **a**, **e**, **f** n = 3 biologically independent samples. In **b**, **c**, n = 3 independent experiments. In **g** n = 5 independent experiments. Source data are provided as a Source Data file.

an outgroup. These accessions mainly included widely planted temperate *japonica* (*GJ-tmp-cul*) and *indica* (*XI-cul*) cultivars from China[30]; tropical *japonica* (*GJ-trp*) and *indica* landrace (*XI-land*) from the 3010 *O. sativa* accessions (3 K panel)[31]; a collection of *japonica* type weedy rice (*GJ-tmp-weedy*) and *indica* type weedy rice (*XI-weedy*)[19]; and NGS sequencing of *Or-IIIa* in this study.

Ten haplotypes were identified based on 16 polymorphic sites of the CDS region with two frameshift mutations, six non-synonymous codons, and eight synonymous codons (Supplementary Data 5). Hap1

(*OsGF14h*^SN9816) and Hap2 (*OsGF14h*^WR04-6) were respectively defined as the partial loss-of-function and functional types based on experimental evidence in the present study. We further found that Hap2 had the highest frequency among all sampled wild rice, *Or-IIIa* (Supplementary Data 6), and was defined in the center of the haplotype network with all ten ecotypes (Fig. 5a); Hap2 was thus predicted as the functional-ancestral type of *OsGF14h*. Hap3, Hap4, Hap5, Hap6, and Hap8 can encode the full-length 14-3-3 protein and also be detected in *Or-IIIa*, which can be defined as the functional-ancestral type. The frameshift

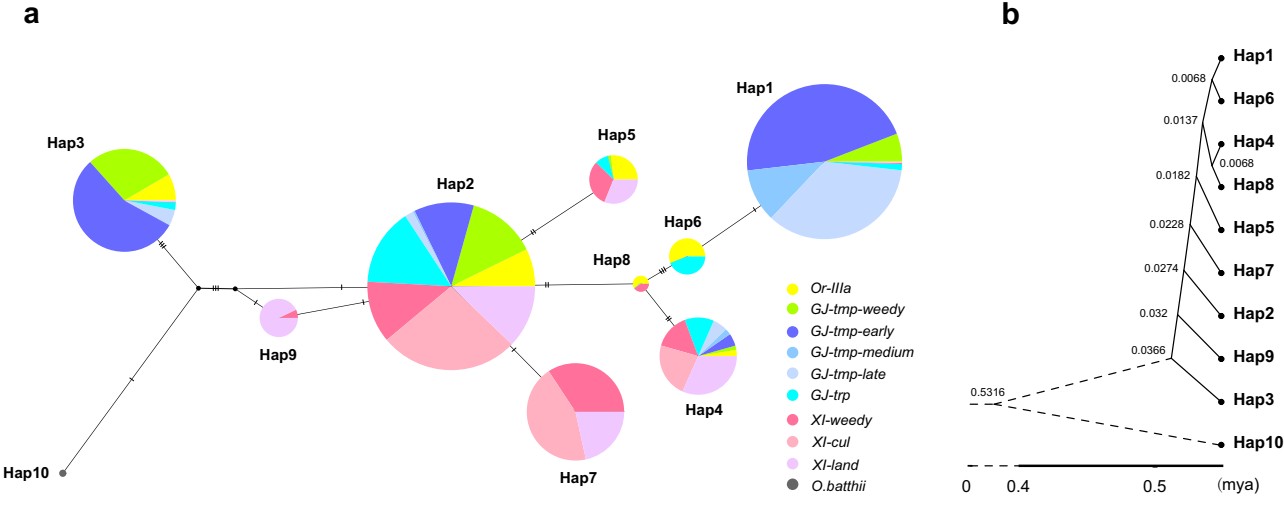

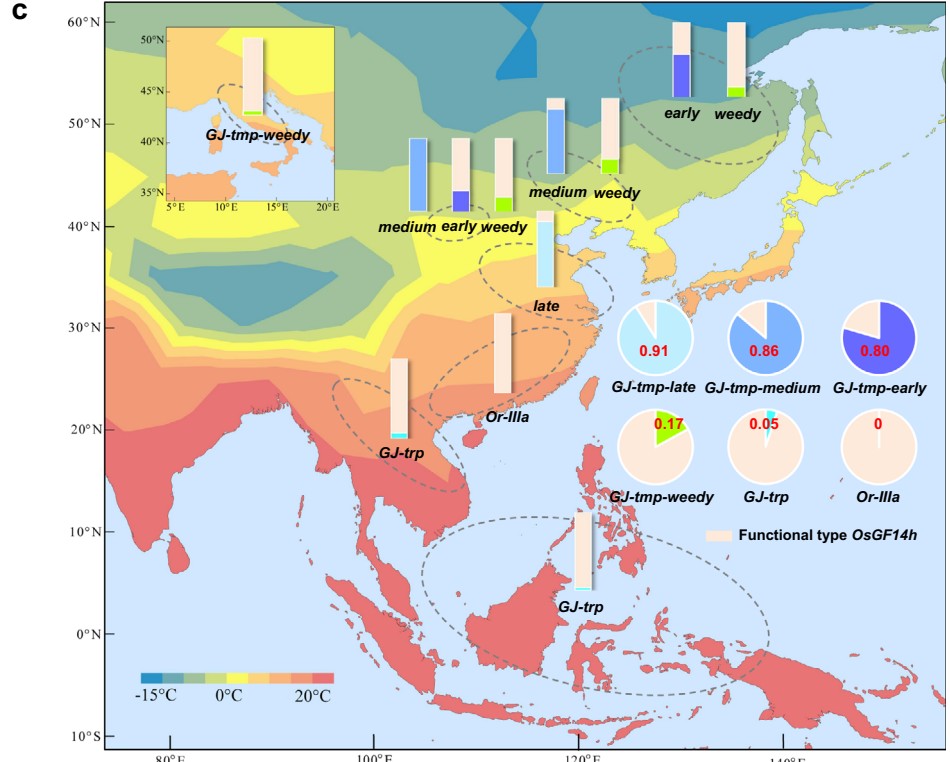

**Fig. 5 | Haplotype network, divergence time estimation, and geographical distribution of *OsGF14h*. a** Haplotype network of *OsGF14h* for ten ecotypes with the *O.bathii* as an outgroup. Circle size is proportional to the sample number for a given haplotype. Black bars on the lines indicate mutational steps between haplotypes. Different colors represent ecotypes, as shown in the illustration. **b** Time-tree is constructed for the ten haplotypes of *OsGF14h* and divergence times presented on the time-tree are estimated by Maximum Likelihood. **c** The origin locations and haplotype frequencies (stacked column chart) for the five *japonica* ecotypes (*n* = 914 biologically independent samples) and the wild ancestors *Or-IIIa* (*n* = 88 biologically independent samples) used in this study (Supplementary Data 7) are marked on this map. Six pie charts represent haplotype frequencies of *OsGF14h* for the 914 accessions in each six ecotypes. The percentage of light pink areas in the stacked column chart and pie chart indicates the frequency of functional type *OsGF14h* in ecotypes to which they belong, and the other color area indicates the frequency of the *OsGF14h*[SN9816] type (indicated by a red number on the pie chart). The heatmap shown on the map depicts the average annual temperature.

mutation of Hap9 caused by the 1 bp insertion led to the loss of four helix in the coding protein resulting in a complete loss of binding sites with OsHOX3. Thus, two frameshift types, Hap1 and Hap9, were defined as partial loss of function haplotypes, mainly detected in temperate *japonica* and *indica* subspecies separately.

There is only one step (the 4 bp deletion) to convert Hap6 to Hap1. However, according to the haplotype network, converting from any other Hap to Hap1 needs more mutational steps. As shown in Fig. 5b,

Hap6 and Hap1 showed the closest divergence time based on a time-tree for the ten haplotypes of *OsGF14h* according to the mcmctree method in PAML software. Based on the above evidence, we infer that Hap1 was likely directly derived from Hap6 (*Or-IIIa* or *GJ-trp*). The emergence of Hap9 may be earlier than Hap1 based on the haplotype network and time-tree. Therefore, we infer that the two partial loss-of-function haplotypes, Hap6 and Hap9, originate independently in the two rice subspecies.

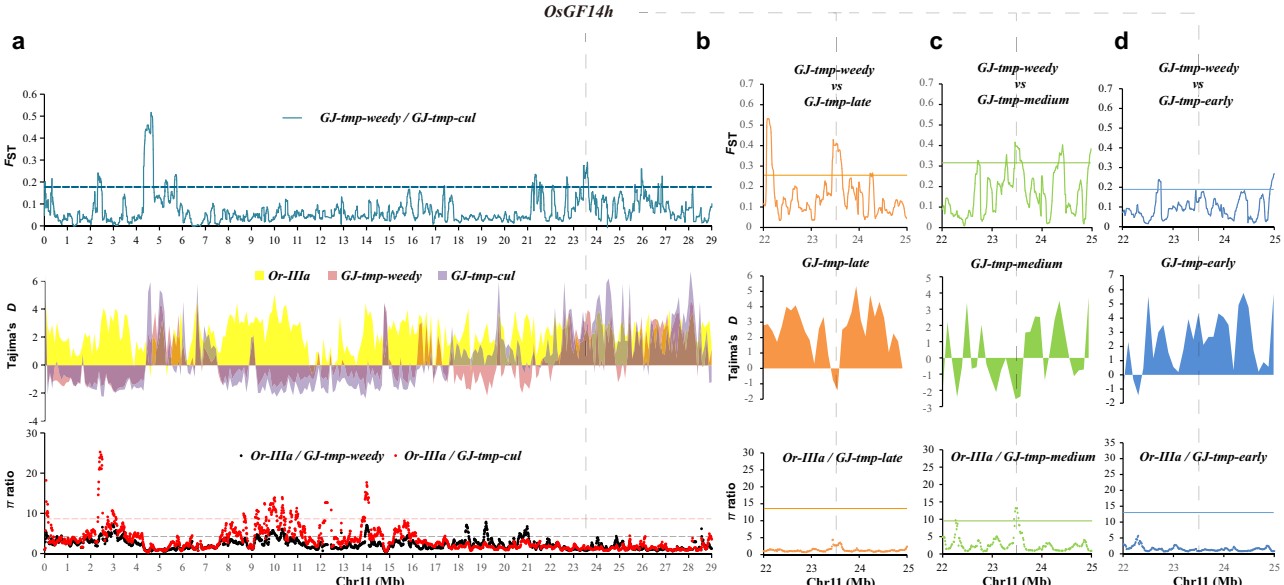

**Fig. 6 | Genomic differentiation and selection between cultivated rice and weedy rice for *OsGF14h* in temperate *japonica* planting areas. a** The top spectrum shows population differentiation between ecotypes of *GJ-tmp-weedy* and *GJ-tmp-cul*. The *y* axis represents 100 kb window-based $F_{ST}$ values. The horizontal dotted line corresponds to a 0.05 significance level of $F_{ST}$ value (0.1842). The middle spectrum shows 500 kb window-based Tajima' *D* values of *GJ-tmp-weedy*, *GJ-tmp-cul*, and *Or-IIIa*, respectively. The bottom spectrum shows selection sweeps defined by 100 kb window-based *π* ratios of *Or-IIIa* divided by *GJ-tmp-weedy* (black dot) and *Or-IIIa* divided by *GJ-tmp-cul* (red dot), respectively. Red and black

horizontal dashed lines correspond to a 0.05 significance level of selection sweep for *GJ-tmp-weedy* (4.83) and *GJ-tmp-cul* (9.11), respectively. **b–d** The spectrums of orange (**b**), green (**c**), and blue (**d**) present 100 kb window-based $F_{ST}$, *π* ratios, and 500 kb window-based Tajima' *D* values, for *GJ-tmp-late*, *GJ-tmp-medium*, and *GJ-tmp-early* on a 3 Mb genomic interval (22 to 25 Mb) of chromosome 11, respectively. Horizontal solid lines correspond to a 0.05 significance level for each spectrum. The *x*-axes of all spectrums represent the physical position of chromosome 11. *OsGF14h* genomic region is indicated with gray dashed lines. Source data are provided as a Source Data file.

We focus on the evolutionary significance of *OsGF14h* in the *japonica* subspecies due to the main functional variation *OsGF14h*[SN9816] occurred mainly in temperate *japonica* planting areas (Fig. 5c). Our results suggest that *OsGF14h* exhibited a significant differentiation signal between *GJ-tmp-weedy* and *GJ-tmp-cul* based on window-based *F*st on chromosome 11 (Fig. 6a), which corresponds to the differentiation of haplotype frequency (Fig. 5c). The haplotype frequencies of *OsGF14h*[SN9816] in three ecotypes of *GJ-tmp-cul* (*GJ-tmp-late* = 0.91, *GJ-tmp-medium* = 0.86, and *GJ-tmp-early* = 0.80) are higher than that of *GJ-tmp-weedy* (0.17). However, window-based Tajima' *D* values (*GJ-tmp-weedy* and *GJ-tmp-cul*) and *π* ratios (*πOr-IIIa* / *πGJ-tmp-weedy* and *πOr-IIIa* / *πGJ-tmp-cul*) do not support the selection and domestication occurred in *OsGF14h* region in both *GJ-tmp-weedy* and *GJ-tmp-cul* (Fig. 6a). We further compared *GJ-tmp-weedy* with the three ecotypes of *GJ-tmp-cul* using a spectrum of the three parameters on chromosome 11 (Supplementary Figs. 8–10). The *OsGF14h* also exhibited population differentiation signals between *GJ-tmp-weedy* and *GJ-tmp-late* (Fig. 6b) and between *GJ-tmp-weedy* and *GJ-tmp-medium* (Fig. 6c) based on window-based *F*st analysis. Both of these differentiation signals overlap the selection signals (window-based Tajima' *D*) of *OsGF14h* in *GJ-tmp-late* and *GJ-tmp-medium*, respectively (Fig. 6b, c). Meanwhile, a selection sweep of *OsGF14h* also is detected in *GJ-tmp-medium* defined by window-based *π* ratio (*πOr-IIIa* /*πGJ-tmp-medium*) (Fig. 6c). Interestingly, the differentiation signal between *GJ-tmp-weedy* and *GJ-tmp-early* in *OsGF14h* genomic region becomes weaker, falling below the 5% significant level (Fig. 6d). Meanwhile, the other two parameters also reject this locus as a selection target (Fig. 6d). It is worth noting that *GJ-tmp-early* is mainly composed of landraces, whereas *GJ-tmp-medium* and *GJ-tmp-late* are modern cultivated varieties in temperate *japonica* planting areas[30]. Under this context, the *OsGF14h* exhibits a strong genetic divergence signal between weedy rice and modern cultivated rice but becomes weakening between weedy rice and landrace, implying that *OsGF14h*[SN9816] plays a vital role in the genetic

improvement of modern *japonica* rice, and the 4 bp deletion could be the major causality.

## Discussion

The characterization of natural variation in AG genes and understanding their mechanisms are essential for the genetic improvement of DSR. At present, two genes with natural variation (*OsTPP7* and *OsCBL10*) that act on α-amylase activity have been reported to be related to AG[5,16]. Nonetheless, the genetic mechanism of AG is still largely unknown. In the present study, we identified an ABA response suppressor gene *OsGF14h* that further broadens our understanding of factors underlying AG and the phytohormones regulatory network. The signaling pathway of ABA on germination and dormancy has been extensively studied in crops[32,33], and the effect of ABA-GA antagonism on seed germination has also been thoroughly researched[15,34,35]. However, the underlying mechanism of seed germination under anaerobic conditions is not fully understood. On the other hand, 14-3-3 proteins are a fascinating and complex protein family that can interact with lots of proteins in both mammals and plants[36,37]. However, the consequences of these interactions are little known. Here we reveal insight into the mechanisms and functions for the coding gene of 14-3-3 protein, particularly decoding the two major natural variants of *OsGF14h* deeply. As shown in Fig. 7, we propose a working model to demonstrate how two natural variations of *OsGF14h* respond to flooding stress during germination. The 14-3-3 proteins are reported to be involved in both GA biosynthetic pathways and GA signaling[38,39]. *OsHOX3* is reported as a positive regulator in GAs biosynthetic pathway in the seedling stage[16], which function is further validated during AG by this study. These results indicate that *OsGF14h*[WR04-6] could balance ABA signaling and GA biosynthesis during anaerobic germination. There may be multiple pathways for the positive regulation of GA by *OsGF14h*, which requires further in-depth study.

Seed dormancy 4 (*Sdr4*) and red pericarp gene *Rc* were reported as two essential seed dormancy genes with positive regulatory effects

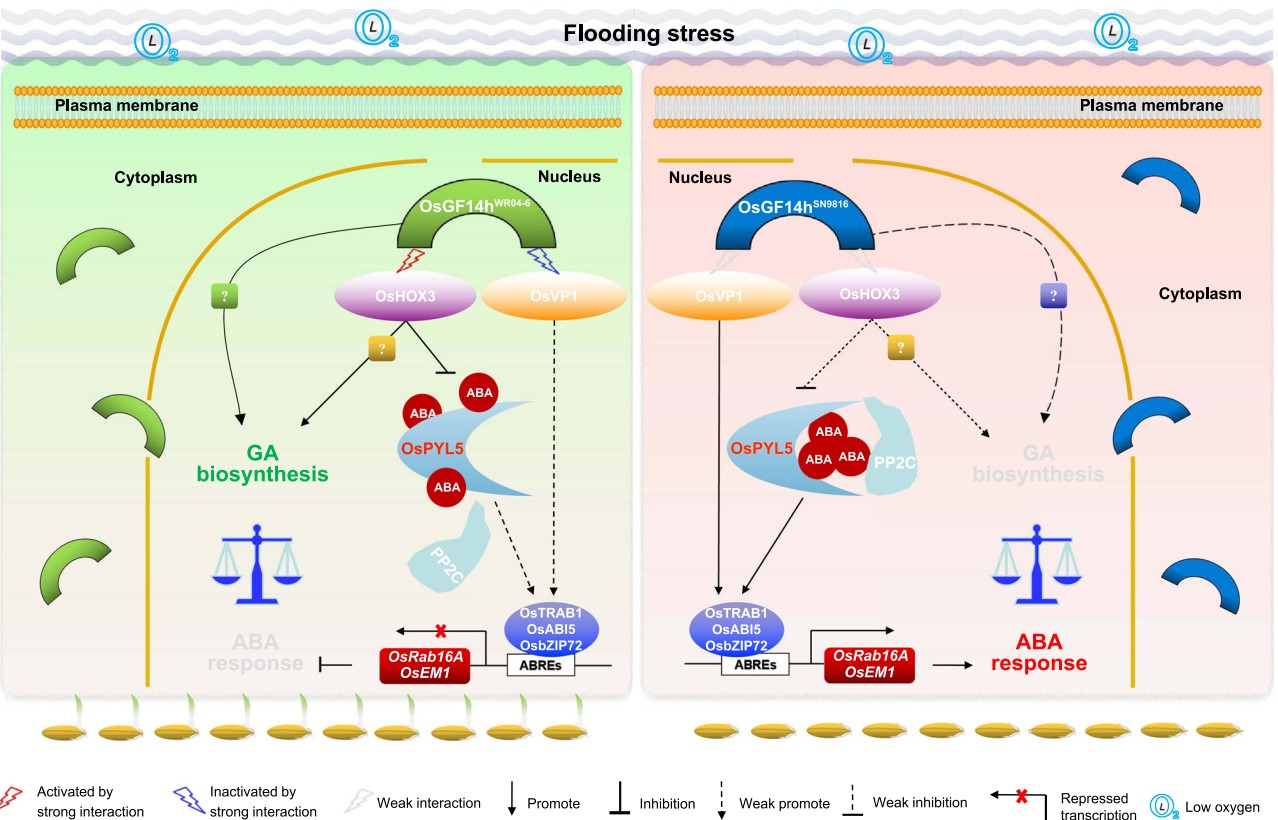

**Fig. 7 | Proposed working model for the regulation of anaerobic germination in rice by *OsGF14h*.** A 14-3-3 protein, Os*GF14h*$^{WR04-6}$ acts as an ABA signal switch that activates a transcription-repressor OsHOX3 and inhibits a transcription activator OsVP1 by the strong interaction. On the one hand, the downstream ABA-regulation TFs and ABA-responsive genes *Rab16A* and *OsEM1* are suppressed due to the ABA receptor OsPYL5 (a core positive regulator of the ABA signal transduction pathway in seed germination and early seedling growth) is inhibited by OsHOX3. On the other hand, ABA-responsive genes are also inhibited due to the functional

weakening of the OsVP1-TRAB1 regulatory unit. Meanwhile, OsGF14h and OsHOX3 can transactivate unknown downstream genes to promote GA biosynthesis. So far, the balance of ABA signaling and GA biosynthesis pathway triggered by OsGF14h$^{WR04-6}$ finally enhanced the anaerobic germination and anaerobic seedling development ability for weedy rice. In contrast, OsGF14h$^{SN9816}$, as a partial loss-of-function 14-3-3 protein, attenuates interaction with OsHOX3 and OsVP1, resulting in the release of inhibition on OsPYL5, whereupon ABA response becomes sensitive and weak germination under the flooding stress.

on ABA that can effectively inhibit pre-harvest sprouting (PHS)[40–42]. A recent study further demonstrates an additive effect of *Sdr4* and *Rc* on seed dormancy[43]. In the present study, we also verified that *Rc* can reduce PHS by knocking out the *Rc* gene in WR04-6 (Supplementary Figs. 11a, b and 12c). The non-dormancy allele, *sdr4-n*, and white peri-carp allele, *rc*, have been nearly fixed in *GJ-tmp-cul* by the artificial selection with the genetic improvement of *japonica* subspecies due to red pericarp and strong dormancy were unfavorable phenotypes (Supplementary Data 7). On the other hand, *GJ-tmp-weedy* requires AG and ASD that are modulated by *OsGF14h*$^{WR04-6}$ to ensure population reproduction by natural selection. However, *OsGF14h*$^{WR04-6}$ also increases the PHS risk due to the high expression in the filling and maturity stage due to it inhibiting the seed ABA response (Fig. 2a). In particular weedy rice matures early without environmental limits for the PHS. In the context that *sdr4-n* is also fixed in *GJ-weedy* (Supplementary Data 7), it is ingenious that *Rc*, as a signature gene of weedy rice, may be the perfect one to reduce the risk of *OsGF14h*-induced PHS because not expressed at germination and seedling stages (Supplementary Fig. 11c), and whereby not counteract the ability of AG and ASD that modulated by *OsGF14h*$^{WR04-6}$. Thus, we infer that *OsGF14h* and *Rc* are co-inheritance and together contribute to the differentiation between *GJ-cul-weedy* and *GJ-tmp-cul* by manipulating the balance between ABA and GA under natural and artificial selection (Fig. 8).

The alleles of *SUB1A*, *SK1*, *SK2*, and *SD1*, identified from wild *Oryza* species, are crucial to flooding survival[8], which indicates that they arose in ancestral populations in flooded ecosystems[1]. However, domestication has modified the phytohormones network to achieve the

production needs for cultivated rice. The competition between cultivated rice and co-existing weedy rice revealed in the present study provides a practical example of how the genetic network of phytohormones responds to different selection pressures. De-domestication, semi-domestication, and outcrossing are the leading hypotheses for the origin of weedy rice, which have been proposed and debated for decades[17–19,23,44]. Regardless of how weedy rice originated, the consequence is the birth of new diversity. More importantly, exploring adaptive genes from weedy rice germplasm resources can retrieve superior alleles that were lost during domestication and modern breeding selection, contributing to crop genetic improvement.

## Methods
### Plant materials
A recombinant inbred line (RIL) population was created with 168 lines derived from the cross between weedy rice WR04-6 and AG-sensitive cultivated rice Qishanzhan at Shenyang Agricultural University, China. All RILs and the two parents were selected to detect anaerobic germination (AG) values. A total of 190 accessions of the GWAS panel, including *japonica* weedy rice (*GJ-weedy*), *japonica* landrace (*GJ-tmp-land*), and *japonica* cultivated are (*GJ-tmp-cul*) listed in Supplementary Data 2.

### Evaluation of germination rate and coleoptile length
Seeds of each variety were grown in the field, and their seeds were harvested 45 days after heading, air dried, and stored at 50 °C for 7 days to break dormancy. Three independent biological replicates

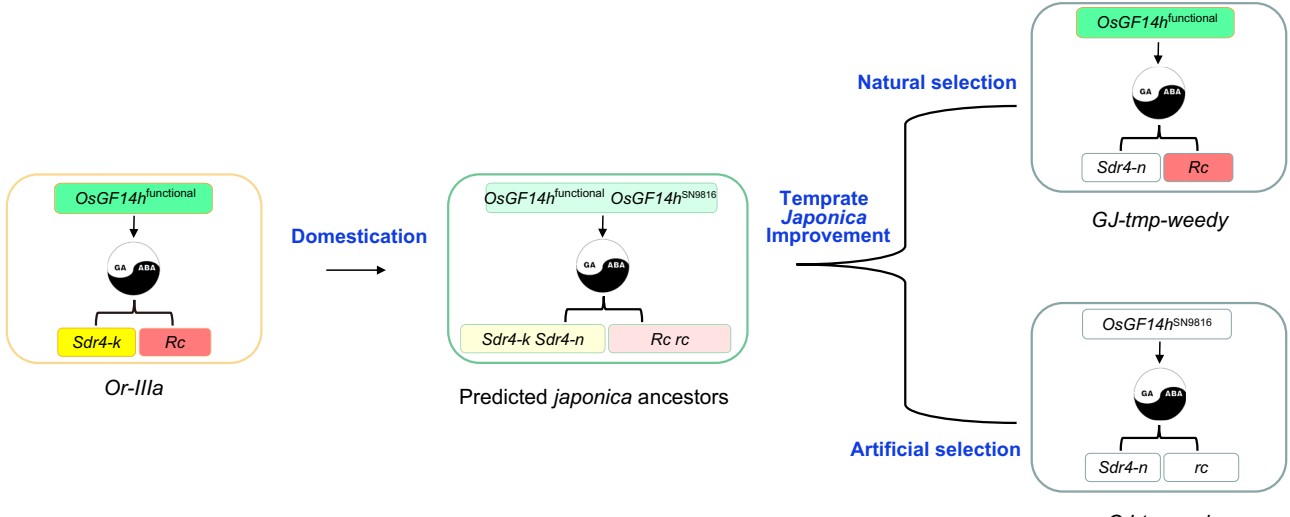

**Fig. 8 | Evolutionary model of OsGF14h in *Japonica* rice.** As an ancestral population, *Oryza rufipogon* (*Or-IIIa*) carries functional alleles of *OsGF14h*, *Sdr4* and *Rc* as a stable unit to maintain robust germination potential for flood adaptation while avoiding pre-harvest sprouting (PHS). Since the red pericarp and strong dormancy are unfavorable traits in the temperate *japonica* planting area, the non-dormancy allele (*Sdr4*-n) and white pericarp allele (*rc*) are gradually fixed in *GJ-tmp-cul* by the artificial selection with the genetic improvement of *japonica* subspecies. Without the double protection from *Sdr4-k* and *Rc*, the *OsGF14h*^SN9816 allele was selected to avoid PHS risk. On the other hand, *GJ-tmp-weedy* requires *OsGF14h*^WR04-6 to enhance anaerobic germination and anaerobic seedling development, ensuring a competitive advantage with cultivated rice for population reproduction by natural selection. In the context that *Sdr4*-n is already fixed in *GJ-tmp-weedy*, *Rc* co-inheritance with functional Os*GF14h* and retained in the weedy rice population. Essentially, the dynamic balance between GA and ABA is the protagonist in this evolutionary story.

with 30 seeds per replicate were then sterilized with 1.5% (v/v) sodium hypochlorite and subsequently incubated in a 50 ml glass (30 mm × 100 mm) at 30 °C under dark conditions for four days under anaerobic conditions. Seeds were considered germinated when white embryo protrusion was visible. The mean germination rates (%) of each variety were recorded after 72 h of treatment for GWAS. The mean coleoptile lengths of each variety were measured on the fourth day.

### QTL mapping
The germination level for each line of the 168 RILs after 72 h of anaerobic treatment was used to determine the QTL phenotype in three independent biological replicates with 30 seeds per replicate. Lines that can germinate and develop coleoptile like the weed rice parent, WR04-6, was defined as 1; lines that cannot germinate like the cultivated rice parent, Qishanzhan, were defined as 0; and lines that can germinate but no obvious coleoptile formation were defined as 0.5 (Supplementary Data 1). For QTL genotyping, we used the previously reported genetic map to calculate QTLs[22] based on inclusive composite interval mapping (ICIM) methods in IciMapping software (Version 4.1)[45].

### DNA sequencing and SNP calling
In the GWAS panel, SNPs of the 190 accessions were obtained by NGS sequencing and re-calling fastq files from published NGS database[23,31,46,47]. First, paired-end sequence data of 114 accessions, including 36 temperate *japonica* weedy rice, 65 temperate *japonica* cultivars, and 13 temperate *japonica* landraces, were generated by Illumina HiSeq4000 covering an average approximate depth of 23.2x for each sample. Then, these fastq files were integrated with published fastq files of 76 *japonica* samples for joint-calling of SNPs and Indels (Supplementary Data 2). The raw paired-end reads were first filtered into clean data using fastp software (Version 0.20.1)[48] with default settings. Clean reads of each accession were mapped to the rice reference genome Os-Nipponbare-Reference-IRGSP-1.0 pseudomolecules using BWA software (Version 0.7.18)[49]. Then the variants detection were applied by Samtools software (Version 1.12)[50] and GATK (Version 4.2) with the HaplotypeCaller program for joint genotyping[51].

The SNPs and indels set detected in the above process was filtered according to a threshold; –min-alleles 2 –max-alleles 2 –maf 0.05 –max-missing 0.8 –minDP 3 –minQ 30, to reduce the variants false discovery rate using vcftools ver3.0 software. Finally, generating 1,160,454 SNPs and 150,990 indels were subjected to principal components analysis (PCA) based on Plink software (Version 1.9)[52].

### GWAS and candidate gene detection
For GWAS, we first performed imputation for the genotype of 190 samples using BEAGLE ver4.0[53]. Then the analysis was performed using EMMAX software (Version emmax-intel-binary-20120210)[54] by fitting a linear mixed model with 1,311,444 genetic markers (MAF = 0.05) and the first two PCs from PCA as covariates to control population structure. The threshold for genome-wide significance was determined by Bonferroni correction, and the significance level was determined as $8.12 \times 10^{-7}$ (0.05 significance level) and $1.62 \times 10^{-7}$ (0.01 significance level). Visualization of GWAS results (manhattan plot and QQ plot) and linkage disequilibrium block were performed using CMplot package (Version 4.0) in R software (Version x64 4.1.1) and LDBlockShow software (Version 1.36)[55]. The preselection of causal haplotype among candidate genes annotated by MSU Rice Genome Annotation Project Release 7 was conducted using CandiHap package[56] of R software (Version x64 4.1.1) by providing the gff file downloaded from http://rice.uga.edu./downloads.shtml. The haplotype of each candidate gene was defined by all detected SNPs and indels within a region of 2000 bp upstream to 500 bp downstream of annotated genes according to the default setting of the CandiHap package of R x64 4.0.1 software. We then conducted expression analysis for seven candidate genes from the seeds of weedy rice WR04-6 after 72-hour anaerobic treatment by qRT-PCR. BLUP analyses were performed based on a subset of SNPs ($n = 54,227$) that were randomly selected from the GWAS markers using the Sommer package (Version 4.1.7) in R software (Version x64 4.1.1)[57].

### Evolutionary analysis of *OsGF14h*
Haplotype frequency analysis for each rice ecotype in the *OsGF14h*^WR04-6 coding region was calculated using DnaSP software (Version 5)[58]. A

haplotype network was built based on 16 mutation sites of the CDS region for ten ecotypes with the *O.bathii* as an outgroup based on the 'Minimum Spanning Network' approach implemented in PopART software (Version 1.7)[59]. The geographic map with average annual temperature was plotted using ArcGIS software (Version 10.6). Time-tree is constructed for the ten haplotypes of *OsGF14h* with the *O.bathii* as an outgroup according to MCMCTREE implemented in PAML software (Version 4.5). Selection sweeps, 100 kb window-based Tajima' *D* and $F_{ST}$ were calculated using vcftools software (Version 3.0)[60].

## Plant transformation

To generate the overexpression constructs, the full-length CDS of *OsGF14h*[WR04-6], *OsGF14h*[SN9816], *OsGA20ox1*, and *OsVP1* from WR04-6 were amplified and then cloned into pCAMBIA1301S with a CaMV 35 S promoter to generate *p35S:OsGF14h-GFP*, *p-35S:OsGA20ox1-GFP*, and *p-35S:OsVP1-GFP* constructs, respectively. To generate the knockout vectors, the specific sgRNA targeting *OsGF14h*, *OsGA20ox1*, *OsVP1*, and *Rc* were designed online (http://crispr.hzau.edu.cn/CRISPR2/). *Agrobacterium tumefaciens* strain EHA105 carrying the gene knockout vectors were used to infect callus tissue induced from the weedy rice WR04-6. Hygromycin-containing medium was used to select hygromycin-resistant callus, and then, the callus were transferred to regeneration media to generate green plants[61]. Plant transformation for *OsGA20ox1* and *Rc* were by Biorun Bioscience Co., Ltd. Wuhan, China. The mutation sites of the *gf14h*, *ga20ox1*, and *rc* mutants are shown in Supplemental Fig. 12. *Agrobacterium*-mediated transformation was used to generate transgenic rice plants. Primer sequences are listed in Supplementary Data 8.

## Real-time PCR

Different tissues and organs of rice were excised from different plants and immediately put into liquid nitrogen for RNA extraction. Total RNA was prepared using the RNAiso Plus Reagent (Takara Bio Inc, Dalian, China) according to the manufacturer's instructions. cDNA was obtained with PrimeScript™ RT Master Mix (Takara Bio Inc, Dalian, China). For qRT-PCR, the cDNA was mixed with SYBR Premix Ex Taq II (TAKARA, Tokyo, Japan) and analyzed using an Applied Biosystems QuantStudio 3 system (Thermo Fisher Scientific, USA).

## Subcellular localization

Rice protoplasts were extracted from a 7-day-old leaf sheath of Nipponbare, which was cultured in the dark. The fusion construct pCAMBIA1301S-35S: *OsGF14h*-GFP was co-transformed into rice protoplasts with 35S::NLS::CFP, used as a nuclear marker. The constructs were introduced into protoplasts through polyethylene glycol (PEG) mediated transformation[62,63]. After incubation in the dark for 24–48 h, the subcellular distribution of GFP fluorescence was determined using confocal microscopy (Nikon C2-ER, Japan). Primer sequences are listed in Supplementary Data 8.

## Germination test by exogenous application of ABA

A total of 35 seeds of WT and knockout line (*gf14h*) were anaerobic-incubated at 30 °C with a gradient concentration of ABA solution as treatment and with pure water as control, respectively. The concentration of ABA solution was set to 0, 0.1, 1, 5, 10, and 50 μM, respectively. The germinated rate was recorded at 72 h of incubation.

## Determination of endogenous GA concentration

To measure the endogenous content of GA, the seeds were prepared at 0, 6, 12, and 24 HAI (hours after imbibition) under anaerobic stress. Liquid nitrogen frozen germinated seeds (50 mg fresh weight) were ground into powder and extracted with the traction method (methanol/water/formic acid = 15:4:1, V/V/V). The extracts were vortexed and centrifuged at 4694 × *g* under 4°C for 10 min. The supernatants were dried by evaporation under the flow of nitrogen gas at room

temperature, then dissolved in 200 μl of methanol. The sample extracts were analyzed using an LC-ESI-MS/MS system (HPLC, Shimpack UFLC SHIMADZU CBM30A system; MS, Applied Biosystems 6500 Triple), and the data were analyzed by Metware Biotechnology Co., Ltd. Wuhan, China. Three replicates of each assay were performed.

## Yeast two-hybrid assay

The Matchmaker Gold Yeast Two-Hybrid system (Clontech) was used to perform the yeast two-hybrid (Y2H) screen. Various tissues (including 30-day seedlings and roots, as well as the stems, leaves, young panicles, and seeds at the reproductive stage) were mixed to construct a two-hybrid library. For the Y2H screening, the CDS of *OsGF14h*[WR04-6] was cloned into vector *p*GBKT7 between the *N*deI and *E*coRI sites and functioned as bait. The yeast strain Y2H Gold (Clontech, Cat. No. 630489) was used for transformation. To verify the interaction between (OsGF14h[WR04-6], OsGF14h[SN9816]) and (OsHOX3, OsVP1) and between OsGF14h and OsVP1 in yeast, the CDS of *OsHOX3* and *OsVP1* were cloned into the *p*GADT7 vector using the *N*deI and *E*coRI sites; then, the *p*GBKT7-*OsGF14h*/*p*GADT7-*OsHOX3* and *p*GBKT7-*OsGF14h*/*p*GADT7-*VP1* pairs were separately transformed into yeast strain Y2H Gold and grown on SD/-Leu/-Trp medium. The positive transformants were further detected on SD/-Leu-Trp-His-Ade medium as described in the Yeast Protocols Handbook (Clotech, PT3024-1). The interaction between pGADT7-T7 and pGBKT7-53, and between pGADT7-T7 and pGBKT7-Lam were regarded as a positive and negative control, respectively. Primers used for construction are listed in Supplementary Data 8.

## Co-immunoprecipitation assay

For the co-immunoprecipitation (co-IP) assay, the full-length CDS of *OsGF14h*[WR04-6], *OsGF14h*[SN9816], *OsHOX3*, and *OsVP1* were cloned into pBWA(V)HS to construct the vectors of pBWA(V)HS-*OsGF14h-GFP*, pBWA(V)HS-*3×FLAG-HOX3*, and pBWA(V)HS-*3×FLAG-VP1*, respectively. The vectors were transformed into *Agrobacterium* strain GV3101, then injected into *Nicotiana benthamiana* leaves. After 48 h, total protein was extracted according to the manufacturer's instructions, then incubated with GFP-Trap coupled to agarose beads (TransGen, DP501-01) for 2 h, then washed three times with buffer (10 mM Tris-HCl, pH 8.0, 150 mM NaCl, 0.5 mM EDTA, 2 mM DTT, 0.1% NP-40). 5×SDS loading buffer was added to the immunoprecipitated proteins and denatured at 95 °C for 10 min and resolved on 10% acrylamide gels. Individual bands were detected using Supersignal West Pico Chemiluminescent Substrate (Thermo) and the ChemDoc™ Touch Imaging system (Bio-Rad). GFP protein was used as a negative control in each set of experiments. The dilution for anti-FLAG (MBL, M185-3L) and anti-GFP (TransGen, HT801-02) antibodies was 1:5000. Primer sequences are listed in Supplementary Data 8.

## Pull-down assay

The CDS of *OsHOX3* or *OsVP1* was cloned into pGEX4T-1 to construct GST-OsHOX3 or GST-OsVP1 fusion protein, and the CDS of *OsGF14h*[WR04-6] or *OsGF14h*[SN9816] was inserted into pET28a to construct and His-OsGF14h[WR04-6] or His-OsGF14h[SN9816], which were expressed in *Escherichia coli* Rosetta (DE3). The recombinant proteins were induced by 0.5 mM isopropyl-ᴅ-thiogalactopyranoside (IPTG) and purified using Ni NTA beads or GST beads (ProbeGene, Xuzhou, China). Then equal quantities of GST-OsHOX3/GST-OsVP1/GST and His-OsGF14h[WR04-6]/His-OsGF14h[SN9816] recombinant proteins were incubated in 1 mL of GST pull-down buffer at 4 °C overnight, and then 50 μL of GST beads were added into the mixture and incubated at 4 °C for 4 h. The beads were washed extensively and boiled. Then the pulled-down proteins were further analyzed by immunoblotting using anti-GST (ABclonal, AE001, 1:5000 dilution) and anti-His (Proteintech Group, 66005-1-Ig, 1:5000 dilution). Primer sequences are listed in Supplementary Data 8.

## Luciferase complementation imaging assay

The luciferase (LUC) complementation imaging assays for the interaction between OsGF14h[WR04-6] and OsHOX3, between OsGF14h[WR04-6] and OsVP1 were performed in *N. benthamiana* leaves. The full-length CDS of *OsGF14h*[WR04-6] and *OsHOX3* and OsVP1 were fused with the pCAMBIA1300-nLuc (pNL) and pCAMBIA1300-cLuc (pCL) plasmids, respectively. The modified plasmids and empty vectors were transformed into *Agrobacterium* strain GV3101. Those cells were resuspended with infiltration buffer (10 mM $MgCl_2$, 100 mM acetosyringone, 10 mM MES, pH 5.6, 1.0 of $OD_{600}$ for each), then equal volumes of *Agrobacterium* suspensions carrying the indicated constructs were co-injected into *N. benthamiana* and cultured at 23 °C for 48 h[64]. The infiltrated leaves were sprayed with 1 mM luciferin in darkness for 5 min and LUC activity was analyzed for 48 h after infiltration using chemiluminescence imaging (Tannon 5200). Each plasmid combination was performed in three independent transformations. Primer sequences are listed in Supplementary Data 8.

## Yeast one-hybrid assay

Yeast one-hybrid (Y1H) assays were performed using the Matchmaker™ Gold Yeast One-Hybrid System (Clontech). First, the full-length CDS of *OsHOX3* was fused with the activation domain of GAL4 protein in the pGADT7 vector as prey. For the construction of bait plasmid, 3×DNA wild and mutated fragment from −1970 to −1948 harboring HD-ZIP3 motif from *OsPYL5* promoter was inserted into the pAbAi vector to construct the bait. After linearizing, the pAbAi bait plasmid was transformed into the Y1H Gold yeast strain, and the minimal inhibitory concentration of aureobasidin A (AbA) of the bait strain was screened. The prey and empty pGADT7 vector were transformed into the recombinant bait-reporter strain. The interaction between pGADT7-p53 and pAbAi-*p53* was regarded as a positive control. The yeast cells were grown on SD/-Leu culture media with or without AbA for 3–5 days at 30 °C. Primer sequences are shown in Supplementary Data 8.

## Dual-luciferase assay

The full-length CDS of *OsHOX3* were separately cloned into the pGreenII 62-SK vector as an effector. The *OsPYL5* promoter was inserted into the pGreenII0800-LUC vector to generate the reporter. The effector and reporter constructs were separately transferred into *A. tumefaciens* strain GV3101 with the helper plasmid pSoup-p19. The strains were co-transformed into 4-week-old *N. benthamiana* leaves and grown for 2 days at 25 °C. The leaves were collected for the determination of firefly luciferase (LUC) and *Renilla* luciferase (REN) activities using a Dual-Luciferase® reporter assay system (Promega, Madison, WI, USA) with a microplate reader (Infinite M200 Pro; Tecan, Männedorf, Switzerland). The empty pGreenII 62-SK vector was used as a negative control. The REN activity under the control of the CaMV 35 S promoter was used as an internal control, and the LUC activity was normalized to the REN activity. Three independent biological replicates were performed. Primer sequences are shown in Supplementary Data 8.

## Electrophoretic mobility shift assay

The CDS of *OsHOX3* and *OsGF14h*[WR04-6] were expressed in *Escherichia coli* Rosetta (DE3) and purified using Ni NTA or GST resin (ProbeGene, Xuzhou, China). For the probes, a 55 bp fragment containing the HD-ZIP3 element as well as a mutated fragment was synthesized and labeled with biotin at the 3′ end, and the unlabeled fragment was used as a competitor. Protein−DNA binding reactions were performed using a Chemiluminescent EMSA Kit GS009 (Beyotime, Shanghai, China) according to the manufacturer's protocol. A 6% native polyacrylamide gel was used for electrophoresis. After that, the complexes were transferred onto a nylon membrane and cross-linked with a UV-light cross-linker. The chemiluminescence was detected using an Azure Biosystems C500 Imager (USA). The probe sequences are shown in Supplementary Data 8.

## Chromatin immunoprecipitation-qPCR assay

Chromatin immunoprecipitation (ChIP) was performed with a 3×FLAG-tagged OsHOX3 expressed in callus. About 2 g callus from the 3×FLAG-OsHOX3-overexpressing line was collected and cross-linked with 1% formaldehyde for 15 min. Then the samples were sonicated for isolation and lysis of nuclei. Antibodies against FLAG (D6W5B, Cell Signaling Technology, 1:50 dilution) were used for immunoprecipitation, and Protein-A + G-agarose beads were used to pull down the protein−DNA complex. After collection, wash, and elution of the immune complex, the precipitated DNA was reversely cross-linked and used for qPCR analysis[65]. Primers were designed in the promoter regions of *OsPYL5* and listed in Supplementary Data 8. Three biological replicates were measured independently.

## Reporting summary

Further information on research design is available in the Nature Research Reporting Summary linked to this article.

## Data availability

The genetic map for QTL mapping has been released previously[22]. The raw sequencing of RILs are available on NCBI BioProject under the accession number PRJNA513407. The raw sequencing of the GWAS and *Or-IIIa* datasets have been deposited in the Genome Sequence Archive in National Genomics Data Center, China National Center for Bioinformation/Beijing Institute of Genomics, Chinese Academy of Sciences (GSA) under accession number CRA005027. The raw sequencing of transcriptome analysis used in this study have been deposited in GSA database under accession number CRA006828. In order to identify the haplotypes of *OsGF14h* and *Sdr4* for GJ-weedy and XI-weedy, we downloaded short-read sequence data from GenBank under the BioProject accession number PRJNA606132, PRJNA344937, and PRJNA295802. The *OsGF14h* and *Sdr4* sequences of widely planted *GJ-tmp* and *XI-cul* cultivars from China were downloaded from MBKbase Database [http://www.mbkbase.org/rice][30]. *OsGF14h* and *Sdr4* sequence data of *XI-land* and *GJ-trp* were downloaded from RiceVarMap v2.0 Database [http://ricevarmap.ncpgr.cn/]. Source data are provided with this paper.

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

## Acknowledgements

*OsHOX3* loss-of-function mutant *hox3* was provided by Prof. Jianmin Wan from the Chinese Academy of Agricultural Sciences. The leaves of *Oryza rufipogon* (Or-IIIa) used for NGS sequencing were obtained from National Nursery of Wild Rice Germplasm (Guangzhou), Guangdong Academy of Agricultural Sciences; Rice research institute, Guangxi academy of agricultural sciences; Rice research institute, Fujian Academy of Agricultural Sciences. We thank Prof. Yuanhu Xuan from Shenyang Agricultural University, China, for suggestions on the molecular biology Experiment. We thank Dr. Baolei Jia from Chung-Ang University, South Korea, for help in the prediction of protein structure. We thank Prof. Chengzhi Liang from Chinese Academy of Science, for help in the genotyping. This work was supported by the National Natural Science Foundation of China (Grant Agreement No. U1708231, W.C. and J.S.), the National Key R&D Program of China (Grant Agreement No. 2017YFD0100501, J.S. and Z.X), and the Support Program for Young Scientific and Technological Innovation Talents of Shenyang, China (Grant Agreement No. RC210408, J.S.).

## Author contributions

J.S. designed this study. J.S., G.Z., Y.C.H., X.Z., and R.G. performed AG phenotyping and QTL mapping, analysed the data. J.S. and X.Y. performed the GWAS and molecular evolutionary analysis. G.Z., R.G., and H.L. performed genetic transformation experiments and phytohormones content determination. G.Z., Z.C., X.K., Y.Q.H., and Z.L. conducted biochemistry experiments. J.S., G.Z., Z.C., X.K., Q.X., and L.T. wrote the manuscript with contributions from all author. W.C., D.M., Z.X., and J.S. supervised the project.

## Competing interests

The authors declare no competing interests.
