## [Peer Review File · Nature Communications]

Regain flood adaptation in rice through a 14-3-3 protein
OsGF14hReviewers' Comments:

Reviewer #1:

Remarks to the Author:

This manuscript presents the identification of a 14-3-3 protein OsGF14h positioned on chromosome 11, which was detected in both bi-parental mapping and GWAS, controlling the anaerobic germination trait. This gene boosts anaerobic germination tolerance by acting as a signal switch to inhibit ABA signaling and enhance GA synthesis via interactions with the OsHOX3 and OsVP1 transcription factors. The authors also claimed that OsGF14h was co-domesticated with Rc (red pericarp gene on chromosome 7) to promote divergence between two subgroups of temperate japonica rice, cultivated vs. weedy, through both artificial and natural selection.

This is an excellent study, which not only revealed the role of 14-3-3 protein OsGF14h regulator, but also identified and confirmed its direct interactions with two downstream transcription factors, OsHOX3 and OsVP1. Further, the authors also uncover the domestication of this gene along with other key domestication genes, Sdr and Rc. Ultimately, OsGF14h can be applied to improve anaerobic conditions during germination and early seedling growth, especially for temperate japonica rice and other cultivated rice varieties that do not possess the tolerant alleles.

I don't have any major concerns with this manuscript. But the following are my minor comments to improve the clarification of the manuscript:

- P4, line 78-79: Please mention the parental name and IDs here.
- P4, line 82: What is the R2 value (phenotypic variant explained)?
- P4, line 83-84: Please clarify here if GJ-tmp is "temperate japonica" group? Is the GJ-weedy referring to the "japonica weedy" rice subgroup? It is necessary to introduce the term when it is first mentioned here.

Please also describe clearly each subgroup mentioned in Extended Table 1 as footnotes.

- The figures need to be mentioned in order (for example, Fig. 1c was mentioned after Fig. 1a (not after Fig. 1b)—line 88). Please double-check throughout the manuscript.
- In Extended Data Fig. 1g, LOC_Os11g39370 had a relatively high expression. In this case, the authors need to explain why this gene was not considered as a candidate as well.
- Fig. 1b: Please fix the chromosome number in the X-axis (it was shifted).
- Fig. 1e: Please mark where is the OsGF14h position (which blue bar?)
- Fig. 1f: It is not clear here what the differences are among the three haplotypes mentioned—needs more details here.
- P9, line 180: it is supposed to be noted here that in Fig. 3i OsbZIP72, the relative expression between WT and gf14h-15 was insignificant (even though there is a slight difference).
- P10, line 10: "...present (Fig. 4d, e, and f)" change to "...present in anaerobic germination conditions (Fig. 4d, e, and f)"
- Fig 6b: I assume the gf14h are homozygous already in the GJ-tmp-cul (majority of this subgroup has this allele)? Currently, it was depicted in a heterozygous state.
- Add footnotes for the rice subgroups (Extended Table 4)
- P16, line 334: How many seeds were used per rep, and how many reps were used for both QTL mapping and GWAS?
- P16, lines 340-343: Why were the actual germination rates of the RILs not used here? (instead categorical numbers (1; 0,5; 0) were used?) I predict this will give a more accurate measurement.
- P22, lines 475-476: What was the reason to perform the LUC complementation in *N. benthamiana* leaves instead of rice leaves? Was it more difficult to detect in the rice leaves?
- Extended Table 5: Please put the name of the rice accessions
- Please double-check typos/English throughout the manuscript: For example, Line 9: "...to promoted..." change "...to promote..."; Fig 2b: Change "Thecomparison..." to "The comparison..."; Fig 6b: "...have allfunctional..." change to "...have all functional..."; line 426: "35 seeds..." change to "A total of 35 seeds..."; etc.

Reviewer #2:

Remarks to the Author:

This ms by Sun et al. identified a 14-3-3 protein-coding gene OsGF14h from QTL mapping and GWAS that confers tolerance to anaerobic stress in weedy rice. They found that OsGF14h inhibited ABA signaling and enhanced GA synthesis by interacting with the transcription factors OsHOX3 and OsVP1. The manuscript reported some intriguing components; however, the current study is preliminary. More molecular and genetic evidence is needed to support their conclusion and final working model. Most importantly, how the natural variation of OsGF14h affects its function should be shown and discussed. Also, the link of OsGF14h with ABA/GA pathway lacks genetic evidence.

1. Line108, the author demonstrated that "SN9816 has a total of six polymorphic sites in the coding region compared with WR04-6" and proposed that "This natural variation of OsGF14h may have a genetic effect on AG tolerance." As a positive regulator of rice anaerobic tolerance, how does OsGF14h respond to anaerobic conditions, and how do these polymorphisms affect its function during anaerobic stress response?
2. The interaction evidence of OsGF14h with OsHOX3 and OsVP1 should be further strengthened. The quality of Fig3b and Fig3d should be improved. Most importantly, the authors should dissect what's the biological significance of their interaction and whether the natural variation of OsGF14h affects these interaction.
3. Fig3d-e, the ChIP experiment should be provided to verify the direct binding and regulation of PLY5 promoter by OsHOX3.
4. Fig3g-i and Fig4a-b, the pattern of ABA and GA response gene expression should be detected in WT, OE (at least two independent lines) and mutant under normal and stress conditions. This will help to determine whether OsGF14h is indeed involved in the signal transduction and genotype x treatment effect.
5. The statistical methods should be corrected or added in Fig3e and Fig3g.
6. The working model is quite preliminary and inaccurate. For instance, what does "functional OsGF14h" mean? The link between OsHOX3 and GA signaling is lacking; and the inhibitory function of OsGF14h on OsVP1 has not been revealed. Overall, the model should be modified carefully.

Reviewer #3:

Remarks to the Author:

The present study revealed that OsGF14h alleles in rice are a major causality to control the variation of anaerobic germination and seedling development using genetic mapping and molecular biology experiments. OsGF14h is involved in crosstalk between ABA and GA signaling pathways. As the optimization of GA signaling for rice cultivars is well known, it makes sense that most japonica lines have less sensitive GA signaling by the loss-of-function OsGF14h, and the findings will largely contribute to rice breeding.

Overall, QTL and GWAS mappings are mostly fine, but the evolutionary analysis is not qualified, and there is too much speculation. Also, many details are missing in this manuscript. Please spell out all abbreviations, including gene names, and explain how each statistic and result lead to each

conclusion.

Major comments

About QTL mapping, materials and methods are insufficient. According to L79, the authors CONSTRUCT RILs derived from WR04-6 and Qishanzhan, but I guess they used F8 RILS generated in their previous work (Sun et al., 2019, Mol Plant). Therefore, they could skip genotyping of RILs, and the genetic map was available. Please describe the methodology accurately as readers can follow it. Also, how many RILS were used for this mapping?

L205, for "Selection and population differentiation driven by a 4bp deletion in OsGF14h", there is no evidence to support this claim. Only the thing the data can tell is, "OsGF14h is a promising candidate involved in domestication, and the 4bp deletion could be the major causality." Many polymorphisms are linked with the 4bp deletion, and the effect of each polymorphism cannot be evaluated easily. Therefore, rescue experiments using the entire coding region, including other polymorphisms (Fig 2b), are not sufficient to claim the effect of the deletion technically.

L227-228, the haplotype analysis is not informative to clarify the allele history, and I do not understand why the non-functional allele seems to be derived from Hap6. If the authors want to discuss this, simulation studies must be helpful to estimate allele age and differentiation timing of the cultivars. Maybe the authors can put Fig 5 and demonstrate the geographic distribution of alleles with the climate data.

L231, do the authors mean "the Pi ratios suggested signatures of selective sweep on GF14h"? Please indicate the evidence to claim "there is selection." Also, only the pi ratio is too weak to claim the selection. How about Tajima's D on Chr 11? The following article would help to look for selection signatures.

Ref. Li, M., Tian, S., Jin, L. et al. Genomic analyses identify distinct patterns of selection in domesticated pigs and Tibetan wild boars. Nat Genet 45, 1431–1438 (2013)

L242-243, this conclusion is too much speculation. I do not see any evidence to lead this conclusion.

Regarding the connection between OsGF14h and GA synthesis, I think starting with OsGA20ox1 is a kind of cherry-picking way. Sdr4-k and Rc are the same. How is RNA-seq for gf14h to look at genome-wide effects of OsGF14h? Since the authors have generated the transgenic lines, the experiments will not be difficult and time-consuming. Showing overviews of the gene function will strengthen the present study.

Minor comments

L221, Fig. 4a -> Fig. 5a

L231, Fig. 5b -> Fig. 5e?

About GWAS, which minor allele frequency did the authors use? Why were the first four PCs chosen for correction of the population structure? Were indels also used as the genetic markers?

For Fig. 1-a, b, the Manhattan plots' labels do not correspond to the correct positions.

For Fig. 1-b & d, does $-\lg(p\text{-value})$ mean $-\log_{10}(p\text{-value})$?

Please use BULP for Fig1-c and f to show the corrected allelic effects.

L95, how did the authors define haplotype? Please write down the methodology.

L98-102, what is the evidence that the authors concluded: "Os11g39540 is the strongest response to anaerobic treatment"? The methods and the explanation are insufficient. For Extended Data Fig. 1-g, if the y-axis shows the relative values to the control samples, 10% of the difference is not reliable due to the technical limitation of qRT-PCR. Also, the "relative expression level" of Os11g39370 is almost the same level with Os11g39540. I think this is not a big issue because the authors showed the gene effect by experiments in the following section, but clear explanations are still needed.

L112, does OsGF14h of SN9816 bring complete loss-of-function? For natural variation, frameshift and stop codon after the functional domain do not always loss-of-function. I am curious the 4bp deletion truly brings a loss of functionality?

Reviewer #4:

Remarks to the Author:

In the present manuscript, authors made an effort to identify novel regulators of waterlogging tolerance related to anaerobic germination (AG), particularly important in direct seeding rice cultivation practice. As a result, LOC_Os11g329540 associated to gene OsGF14h, a 14-3-3 protein has been identified. Authors claim that this protein acts as a hub influencing ABA and GA signaling, involved in rice stem elongation in response to submergence and in the development of other morphological adaptation to waterlogging. To this respect, authors report that OsGF14h represses ABA signaling through its interaction with OsHOX2 and OsVP1 which repress the ABA response, potentially at the level of the ABA receptor OsPYL5. OsGF14h overexpressors showed enhanced coleoptile growth and survival rate under anaerobic conditions, but no effect on germination in the presence of ABA has been shown. Conversely, knockouts of OsGF14h showed enhanced sensitivity to ABA in a germination assay in the presence of the hormone. In addition, OsGF14h seems to co-express with OsHOX3 and also physically interact as a result of a Y2H analysis. To this respect, OsHOX3 acts as a OsPYL5 transcriptional repressor binding to its promoter (as a result of a gel shift assay) also under normal conditions. Under anaerobic conditions expression of ABA-dependent genes is associated to the presence of an active OsGF14h. On the contrary, OsVP1 overexpressors exhibited an opposite behaviour to that of OsHOX3 overexpressors potentially indicating that these two genes might have contrasting roles, despite the actual role of OsVP1 has not been further investigated. Taken together, this part seems pretty convincing and the role of OsGF14h as a switch balancing anaerobic growth through manipulation of ABA and GA signaling is plausible. However, the actual role of OsVP1 is less studied and its involvement in the regulation of anaerobic germination in relation to OsGF14h and in balance with OsHOX3 is partially addressed.

1.- Moreover, ABA levels along with GA levels, need to be provided in OsGF14h OE and KO seeds and seedlings.

2.- Authors did not measure bioactive gibberellin levels: GA1, GA3, GA4 or GA7 and that is necessary to link with the physiological/morphological response. Despite it makes sense that *gf14h* plants show reduced GA20 albeit enhanced GA53 levels, bioactive GA levels are required to assess their impact on the physiological response.

3.- Germination assays of OsGF14h OE in the presence of ABA are missing.

The rest of the work trying to link functional OsGF14h alleles with different germplasm accessions is less clear and needs some clarification.

First, this part seems unconnected to the rest of the work, which is primarily focused on the characterization of the role of OsGF14h in the anaerobic germination of rice. I understand that this is part of a bigger project aimed at identifying QTLs related to enhanced aerobic germination in rice and

interesting germplasm resources to introgress this trait into elite varieties. And that, of course, requires other studies to investigate the co-inheritance of other important quality traits (e.g. Sdr4-n and Rc).

The mode of inheritance of functional OsGF14h, absent in SN9816 and Hap1 ecotypes, is rather speculative and requires the survey of more samples as the explanation on how functional OsGF14h found in Hap2 has rendered a non-functional allele in Hap1 is too speculative. To This respect, there is only one haplotype harboring a non-functional OsGF14h allele (Hap1), what happened to Hap9, also with a non-functional allele? This could clarify the mode of inheritance of that non-functional allele. The population differentiation index (F_{st}) is poorly explained. It is not clear what it means in this context: Is it that most SNPs are equally present with no clear bias and those close to 5 and 25 Mb are fixed in the population or come from one of the genotypes? What implications have the different threshold values in this population? Moreover, in the graphs, what is that gap between 20 and 25 Mbp? Is that the centromere? If that is the case that could have implications regarding the recombination of LOC_11g39540.

In the final part of the manuscript, the co-inheritance of Sdr4-n and Rc genes, related to seed quality is interesting from the breeding point of view, but, in my humble opinion, out of the scope of the manuscript and poorly addressed as it stands. Moreover, the association of Rc with ABA signaling through OsVP1 (which has also been poorly investigated in this work) requires more research to decipher the potential interaction with the rest of elements characterized, if any, and their interaction mechanism. For instance, do Rc seeds have differential ABA sensitivity in germination assays than non-Rc seeds? This is probably part of another, bigger, piece of research.

Minor aspects

Figure 1a, chromosome position seems to be displaced, please amend.

L221 Fig. 5a not Fig. 4a.

L255-256 the sentence is confusing, please rephrase. E.g.: [...] resulting in enhanced PHS,

Reviewer #5:

Remarks to the Author:

Jian Sun and his colleagues developed a cultivated-weedy rice RIL population and map-based cloned a functional gene for flood adaptation. To my knowledge, this is the first case to find an important gene from weedy rice population, a special genetic resource for rice breeding. Their experiment of functional validation is solid and acceptable. I believe they will find more important genes from the weedy rice population in future.

Minor comments:

1. A paragraph in INTRODUCTION should be added to introduce weedy rice. I guess a lot of reviewers have no idea about it, for example its origin, distribution and population genomic studies recently, etc.
2. In DISCUSSION section about weedy rice resource, they mentioned semi-domestication about weedy rice. I suggest to give more discussion on different views about origin of weedy rice based on the recent studies on weedy rice.

REVIEWER COMMENTS

Reviewer #1 (Remarks to the Author):

This manuscript presents the identification of a 14-3-3 protein OsGF14h positioned on chromosome 11, which was detected in both bi-parental mapping and GWAS, controlling the anaerobic germination trait. This gene boosts anaerobic germination tolerance by acting as a signal switch to inhibit ABA signaling and enhance GA synthesis via interactions with the OsHOX3 and OsVP1 transcription factors. The authors also claimed that OsGF14h was co-domesticated with Rc (red pericarp gene on chromosome 7) to promote divergence between two subgroups of temperate japonica rice, cultivated vs. weedy, through both artificial and natural selection. This is an excellent study, which not only revealed the role of 14-3-3 protein OsGF14h regulator, but also identified and confirmed its direct interactions with two downstream transcription factors, OsHOX3 and OsVP1. Further, the authors also uncover the domestication of this gene along with other key domestication genes, Sdr and Rc. Ultimately, OsGF14h can be applied to improve anaerobic conditions during germination and early seedling growth, especially for temperate japonica rice and other cultivated rice varieties that do not possess the tolerant alleles. I don't have any major concerns with this manuscript. But the following are my minor comments to improve the clarification of the manuscript:

Response: Thanks so much for the valuable comments that greatly improved our study and are very useful for improving readability of the MS.

- P4, line 78-79: Please mention the parental name and IDs here.

Response: We have clarified parental names and IDs here. (Line 87-88)

- P4, line 82: What is the R2 value (phenotypic variant explained)?

Response: The main QTL we detected with a LOD value of 5.65 and with PVE of 15.85. We updated this data in the revised MS. (Line 93-94)

- P4, line 83-84: Please clarify here if GJ-tmp is “temperate japonica” group? Is the GJ-weedy referring to the “japonica weedy” rice subgroup? It is necessary to introduce the term when it is first mentioned here.

Response: Thanks for the suggestion. We have introduced these terms and listed all the abbreviations in the revised MS.

Please also describe clearly each subgroup mentioned in Supplementary Table 1 as footnotes.

Response: Thanks for the suggestion. We have updated this information in Supplementary Data 1.

- The figures need to be mentioned in order (for example, Fig. 1c was mentioned after Fig. 1a (not after Fig. 1b)—line 88). Please double-check throughout the manuscript.

Response: Thanks for pointing this out. We try to balance the figure's order and the logic of the MS. If some cases are challenging to balance, we will communicate with the editorial office to resolve them.

- In Supplementary Fig. 1g, LOC_Os11g39370 had a relatively high expression. In this case, the authors need to explain why this gene was not considered as a candidate as well.

Response: Thanks for the comments, and this concern is reasonable. We found that the *LOC_Os11g39370* is mainly expressed in leaves, whereas our candidate gene *LOC_Os11g39540* is mainly expressed in seeds based on two Spatio-temporal expression databases. Thus the target gene *LOC_Os11g39540* is our only candidate gene. We have updated this information in the Supplementary Fig.2 of the revised MS.

- Fig. 1b: Please fix the chromosome number in the X-axis (it was shifted).

Response: We are sorry for this mistake and have corrected it in the revised MS.

- Fig. 1e: Please mark where is the *OsGF14h* position (which blue bar?)

Response: Thanks for the suggestion. We have highlighted the position bars of SNPs within the *OsGF14h* gene.

- Fig. 1f: It is not clear here what the differences are among the three haplotypes mentioned—needs more details here.

Response: We are sorry for the confusion. This Boxplot is used to nominate the candidate gene of GWAS that is outputted from the Candihap software. We have incorporated it into Supplementary Fig. 1 to avoid confusion with the gene haplotypes.

- P9, line 180: it is supposed to be noted here that in Fig. 3i *OsZIP72*, the relative expression between WT and *gf14h-15* was insignificant (even though there is a slight difference).

Response: Thanks for the reminder. The pattern of ABA and GA response genes expression were redetected among WT, two independent OE, and knockout lines under both normal and stress conditions. We obtained the expected results among these ABA and GA response genes, as shown in Fig. 4a.

- P10, line 10: "...present (Fig. 4d, e, and f)" change to "...present in anaerobic germination conditions (Fig. 4d, e, and f)"

Response: Thank you for your suggestion. We have updated this information in the revised MS.

- Fig 6b: I assume the *gf14h* are homozygous already in the GJ-tmp-cul (majority of this subgroup has this allele)? Currently, it was depicted in a heterozygous state.

Response: We are sorry that the ‘/’ symbol led to this confusion. Yes, the *gf14h* allele is homozygous, and majority in the *GJ-tmp-cul* subgroup, we use OsGF14h^{SN9816} instead of *gf14h*. We used spaces instead of ‘/’ to indicate the coexistence of the two alleles in Fig. 8 of the revised MS.

- Add footnotes for the rice subgroups (Supplementary Table 4)

Response: Thanks for the suggestion. We have updated this information in Supplementary Data 4 of the revised MS.

- P16, line 334: How many seeds were used per rep, and how many reps were used for both QTL mapping and GWAS?

Response: Thanks for the reminder. Three independent biological replicates with 30 seeds per replicate were used to determine phenotype for both QTL and GWAS. We have updated this information. (Line 414 and Line 423)

- P16, lines 340-343: Why were the actual germination rates of the RILs not used here? (instead categorical numbers (1; 0,5; 0) were used?) I predict this will give a more accurate measurement.

Response: Yes, we fully agree with it. However, we found that the anaerobic germination (AG) performance between RILs and GWAS individuals is difference. AG rates are difficult to quantify in RILs due to the out-of-sync between germination and coleoptile development. Some lines germination well, but the coleoptile development stopped. In comparison, we can more easily decide the AG performance for a given RIL is more like a weedy rice parent or a cultivated rice parent. We also obtained better LOD value and PVE using this way than actual germination rates as phenotype.

- P22, lines 475-476: What was the reason to perform the LUC complementation in *N. benthamiana* leaves instead of rice leaves? Was it more difficult to detect in the rice leaves?

Response: *N. benthamiana* leaves at the appropriate age are large and convenient for injection. Multiple groups can be set on a leaf simultaneously, which is a suitable medium for transient expression. The transient expression method mediated by *Agrobacterium tumefaciens* in *N. benthamiana* leaves is recognized as an effective method to detect protein-protein interactions in molecular biology research. This method is simple, efficient, accurate, and reliable and has a short experimental period. Therefore, in plant studies, *N. benthamiana* leaves are widely used for LUC complementation^{1,2}.

- Supplementary Table 5: Please put the name of the rice accessions

Response: We have updated this information in Supplementary Data 5.

- Please double-check typos/English throughout the manuscript: For example, Line 9: “...to promoted...” change “...to promote...”; Fig 2b: Change “Thecomparison...” to

“The comparison...”; Fig 6b: “..have allfunctional...” change to “...have all functional...”; line 426: “35 seeds...” change to “A total of 35 seeds...”; etc.

Response: We are sorry for these mistakes and have corrected them in the revised MS.

Reviewer #2 (Remarks to the Author):

This ms by Sun et al. identified a 14-3-3 protein-coding gene OsGF14h from QTL mapping and GWAS that confers tolerance to anaerobic stress in weedy rice. They found that OsGF14h inhibited ABA signaling and enhanced GA synthesis by interacting with the transcription factors OsHOX3 and OsVP1. The manuscript reported some intriguing components; however, the current study is preliminary. More molecular and genetic evidence is needed to support their conclusion and final working model. Most importantly, how the natural variation of OsGF14h affects its function should be shown and discussed. Also, the link of OsGF14h with ABA/GA pathway lacks genetic evidence.

Response: Thanks so much for the comments that greatly improved our study, and we fully agree with them. We have supplemented and updated several experiments and analyses to demonstrate the critical issue of “how the natural variation of OsGF14h affects its function”. We hope that our new efforts and responses adequately address the concerns.

1. Line108, the author demonstrated that “SN9816 has a total of six polymorphic sites in the coding region compared with WR04-6” and proposed that “This natural variation of OsGF14h may have a genetic effect on AG tolerance.” As a positive regulator of rice anaerobic tolerance, how does OsGF14h respond to anaerobic conditions, and how do these polymorphisms affect its function during anaerobic stress response?

Response 1: In order to provide more genetic evidence for confirming functional differences between the two major haplotypes, OsGF14h^{SN9816} (Hap1) and OsGF14h^{WR04-6} (Hap2), we created two new overexpression lines of SN9816 with their own CDS overexpressed, O_xOsGF14h^{SN9816}-5 and O_xOsGF14h^{SN9816}-11. The AG performance of these two new overexpression lines showed no significant difference with their wild-type SN9816 during the anaerobic stress response. However, the anaerobic tolerance significantly enhanced when introduced the CDS of OsGF14h^{WR04-6} (Hap2) into SN9816 which was driven by the CaMV 35S promoter. Therefore, we believe that OsGF14h^{WR04-6} (Hap2) is a functional allele and OsGF14h^{SN9816} (Hap1) is a partial loss-of-function or non-functional allele.

Based on the prediction by AlphaFold2, the functional-type OsGF14h^{WR04-6} detected from weedy rice WR04-6 encodes a fully functional 14-3-3 protein with nine alpha-helix structures that could form a clamp to bind the N-terminal loop of OsHOX3. The binding is mainly through the hydrophobic interactions between the C-terminus of OsGF14h^{WR04-6} and the N-terminus of OsHOX3. For the OsGF14h^{SN9816} detected mainly from temperate *japonica* cultivated rice. The frameshift mutation caused by the 4bp deletion in the coding region led to the loss of one helix in the C-terminus, weakened the strength of its interaction with OsHOX3, and two non-synonymous substitutions SNPs that appeared before the 4bp deletion were not located at the binding site. The other two synonymous substitution SNPs and one synonymous substitution SNP that appears after the stop codon may not affect the function.

Another non-functional type *GF14h* (Hap9), is a rare allele detected from *indica* subspecies, the frameshift mutation caused by the 1bp insertion led to the loss of four helices that resulted in complete loss of binding sites with OsHOX3.

We then supplemented and updated several experiments to demonstrate the interaction between 2 main nature variations of GF14h with OsHOX3 and OsVP1 and their effects on downstream genes and anaerobic stress response. Please refer to Response 2 for the comment 2.

2. The interaction evidence of OsGF14h with OsHOX3 and OsVP1 should be further strengthened. The quality of Fig3b and Fig3d should be improved. Most importantly, the authors should dissect what's the biological significance of their interaction and whether the natural variation of OsGF14h affects these interaction.

Response 2: In order to reveal whether the natural variation of OsGF14h affects the protein interaction, We supplemented and upgraded Pull-down, Y2H, CoIP, and luciferase (LUC) complementation imaging assays to demonstrate how this natural variation finally affects the anaerobic germination. We first confirmed that both OsGF14h^{SN9816} and OsGF14h^{WR04-6} can interact with OsHOX3 and OsVP1 *in vivo* by using Y2H and transient expression assay (Fig. 3a and b). These interactions can also be verified *ex vivo* based on a pull-down assay (Fig. 3c). Based of the prediction by AlphaFold2, the WR04-6 type OsGF14^{WR04-6} encodes a fully functional 14-3-3 protein that, with nine alpha-helix structures, could form a clamp to bind the N-terminal loop of OsHOX3. For the SN9816 type OsGF14^{SN9816}, the frameshift mutation caused by the 4bp deletion in the coding region led to the loss one helix in the C-terminus that may weaken the strength of its interaction with OsHOX3, and the other two non-synonymous substitution SNPs before the 4bp deletion were not located at the binding site (Supplementary Fig.3a). The co-immunoprecipitation and Y2H assays further proved the interaction strength between OsGF14h^{WR04-6} and OsHOX3, and between OsGF14h^{WR04-6} and OsVP1 were greater than that of OsGF14h^{SN9816} *in vivo* (Fig. 3a and d).

The supplemented ChIP-qPCR assay and upgraded EMSA further verified the binding capacity of OsHOX3 to the *OsPYL5* promoter (encoding ABA receptor) was enhanced by adding OsGF14h^{WR04-6} protein (Fig. 3l). Therefore, we hypothesized that OsGF14h^{WR04-6} inhibited the expression of *OsPYL5* through interaction with OsHOX3, thus inhibiting ABA signaling. We have upgraded these results in the revised MS.

3. Fig3d-e, the ChIP experiment should be provided to verify the direct binding and regulation of PYL5 promoter by OsHOX3.

Response 3: We used the OsHOX3-overexpressed callus to perform ChIP-qPCR. The results showed that enrichment of promoter fragments was high in the overexpression lines compared with that of *actin*, which proved that OsHOX3 binds *OsPYL5* promoter *in vivo* (Fig. 3i).

4. Fig3g-i and Fig4a-b, the pattern of ABA and GA response gene expression should be detected in WT, OE (at least two independent lines) and mutant under normal and

stress conditions. This will help to determine whether OsGF14h is indeed involved in the signal transduction and genotype x treatment effect.

Response 4: Thanks for the good suggestions. The pattern of ABA response genes has already been detected in WT, OE (at least two independent lines), and mutants under normal and stress conditions, as shown in Fig. 4a. Regulation of GA biosynthesis pathway by *OsGF14h* was further confirmed by RNA-seq (Fig. 4d). The *GA20ox1*, which is crucial for rice development at the initial growth stage, was also detected in WT, OE, and mutants under normal and stress conditions, as shown in Fig. 4e. Importantly, these gene expression results are in line with our predictions. We believe these supplemented experiments, according to your suggestion, further proved that OsGF14h is indeed involved in the signal transduction and genotype x treatment effect.

5. The statistical methods should be corrected or added in Fig3e and Fig3g.

Response 5: We apologize for this lack of rigor and have updated both Fig. 3j and Fig. 3k in the revised MS. The statistical method in Fig. 3j and Fig. 3k was Student's *t*-test at 95% confidence level.

6. The working model is quite preliminary and inaccurate. For instance, what does “functional OsGF14h” mean? The link between OsHOX3 and GA signaling is lacking; and the inhibitory function of OsGF14h on OsVP1 has not been revealed. Overall, the model should be modified carefully.

Response 6: Thank you very much for pointing this out, and we agree with you. We gain further insight into the mechanism of *GF14h* with the new experimental results and analysis according to your valuable suggestions.

OsHOX3 has been reported as a positive regulator in GAs biosynthetic pathway in the seedling stage¹⁶. In order to establish a link between the *OsHOX3* and GA during anaerobic stress, we compared the expression of representative GA biosynthesis-related genes between the wild type of *OsHOX3* and its mutant under anaerobic stress. The result was partially validated by qPCR analysis during AG in this study. The results showed the positive regulation of *OsHOX3* on GA biosynthesis was more reflected in the degradation pathway, in addition to *OsGA20ox1* and *KOS4* in the biosynthesis pathway. (Supplementary Fig. 7). Thus, *OsGF14h* may enhance GAs biosynthesis via but not limited to *OsHOX3*, thereby acting as a role in balancing ABA and GA during anaerobic germination.

RNA-seq for *OxOsGF14h*^{WR04-6-8} and *OxOsGF14h*^{SN9816-5} under both anaerobic and aerobic conditions was conducted to evaluate genome-wide effects of *OsGF14h*. Genes involved in GA biosynthesis are crucial for rice development, especially *OsGA20ox1* plays a dominant role in the initial growth stage³. Finally, GA biosynthesis genes and ABA-responsive genes showed expected differential expression (as the Response to your comments 4). It has been reported that 14-3-3 proteins are involved in both the GA biosynthetic pathway and also GA signaling, which also been proved during anaerobic germination by this study^{4,5}. However,

through which genes and proteins do GF14h regulate GA biosynthesis, further research is needed.

We did not provide one critical information in the first draft that 14-3-3 protein can mediate the trans-regulation of the ABA-responsive factor OsEM by OsVP1, and it was added in the revised manuscript⁶. On the other hand, intensive research has proven that OsVP1, as a vital TF, interacts with a bZIP factor, TRAB1, to inhibit germination by positively regulating the response sensitivity of rice to ABA. In the present study, we revealed that the interaction strength of OsGF14h^{WR04-6} with OsVP1 was greater than that of OsGF14h^{SN9816} *in vivo*, and the ABA-responsive genes were suppressed significantly by OsGF14h^{WR04-6} under anaerobic conditions. Based on the above evidence, we may infer that OsGF14h^{WR04-6} inactivates OsVP1 by the interaction.

These results indicate that *OsGF14h*^{WR04-6} could balance ABA signaling and GA biosynthesis during anaerobic germination. We then modified the model carefully based on the new findings (Fig. 7). For the wording of ‘functional OsGF14h’, we replace ‘functional OsGF14h’ and ‘non-functional OsGF14h’ with ‘*OsGF14h*^{WR04-6}’ and ‘*OsGF14h*^{SN9816}’, respectively.

Reviewer #3 (Remarks to the Author):

The present study revealed that OsGF14h alleles in rice are a major causality to control the variation of anaerobic germination and seedling development using genetic mapping and molecular biology experiments. OsGF14h is involved in crosstalk between ABA and GA signaling pathways. As the optimization of GA signaling for rice cultivars is well known, it makes sense that most japonica lines have less sensitive GA signaling by the loss-of-function OsGF14h, and the findings will largely contribute to rice breeding. Overall, QTL and GWAS mappings are mostly fine, but the evolutionary analysis is not qualified, and there is too much speculation. Also, many details are missing in this manuscript. Please spell out all abbreviations, including gene names, and explain how each statistic and result lead to each conclusion.

Response: Thanks so much for the valuable comments that greatly improved our study and are very useful for improving the scientific logic of this MS. We upgraded the evolutionary analysis, and we believe this part has improved a lot under your guidance. Please see the following reply and Line 279-341 of the revised manuscript for details. We added a list with abbreviations and gene names. We also added details and explanations for each statistic in the revised MS.

Major comments

About QTL mapping, materials and methods are insufficient. According to L79, the authors CONSTRUCT RILs derived from WR04-6 and Qishanzhan, but I guess they used F8 RILS generated in their previous work (Sun et al., 2019, Mol Plant). Therefore, they could skip genotyping of RILs, and the genetic map was available. Please describe the methodology accurately as readers can follow it. Also, how many RILS were used for this mapping?

Response: Thanks for the comments. We skip genotyping information of RILs and refer our previous work. We described the methodology accurately in the revised MS. We use 168 RILs for QTL mapping.

L205, for "Selection and population differentiation driven by a 4bp deletion in OsGF14h", there is no evidence to support this claim. Only the thing the data can tell is, "OsGF14h is a promising candidate involved in domestication, and the 4bp deletion could be the major causality." Many polymorphisms are linked with the 4bp deletion, and the effect of each polymorphism cannot be evaluated easily. Therefore, rescue experiments using the entire coding region, including other polymorphisms (Fig 2b), are not sufficient to claim the effect of the deletion technically.

Response: Thank you for pointing out this critical issue. We fully agree with you. We have supplemented and upgraded several experiments and analyses to demonstrate this issue. We hope that our new efforts and responses adequately address the concerns.

In order to provide more genetic evidence for confirming functional differences between the two major haplotypes, OsGF14h^{SN9816} (Hap1) and OsGF14h^{WR04-6} (Hap2), we created two new overexpression lines of SN9816 with their own CDS

overexpressed, OxOsGF14h^{SN9816-5} and OxOsGF14h^{SN9816-11}. The anaerobic germination of these two new overexpression lines showed no significant difference with their wild-type SN9816 during the anaerobic stress response. However, the anaerobic tolerance enhanced much when introduced WR04-6 type *OsGF14h* (Hap2) into SN9816 driven by CaMV 35S promoter. Therefore, we believe that OsGF14h^{WR04-6} (Hap2) is a functional allele and OsGF14h^{SN9816} (Hap1) is a partial loss-of-function or non-functional allele.

Based on the prediction by AlphaFold2, the functional-type OsGF14h^{WR04-6} detected from weedy rice WR04-6 encodes a fully functional 14-3-3 protein that with 9 alpha-helix structure could form a clamp to bind the N-terminal loop of OsHOX3. The binding is mainly through the hydrophobic interactions between C-terminus of OsGF14h^{WR04-6} and N-terminus of OsHOX3. The OsGF14h^{SN9816} was detected mainly from temperate *japonica* cultivated rice, the frameshift mutation caused by the 4bp deletion in the coding region led to loss one helix in the C-terminus that weakened the strength of its interaction with OsHOX3, and two non-synonymous substitution SNPs that appear before the 4bp deletion were not located at the binding site. The other two synonymous substitution SNPs and one synonymous substitution SNP that appears after the stop codon may not affect the function. We then supplemented and upgraded several experiments to demonstrate how this natural variation finally affects the anaerobic germination. Another non-functional type GF14h (Hap9), is a rare allele detected from *indica* subspecies, the frameshift mutation caused by the 1bp insertion led to the loss four helix, which resulted in a complete loss of binding sites with OsHOX3. On the other hand, we found that overexpression of SN9816's own CDS did not enhance its germination ability. Based on the above evidence, we believe that the structural integrity of 14-3-3 protein determines its functional integrity. Considering that OsGF14h^{SN9816} also can interact with OsHOX3 and OsVP1, thus we defined *OsGF14h*^{SN9816} as a partial loss-of-function type in the revised MS.

In order to reveal whether the natural variation of OsGF14h affects the protein interaction, we supplemented and upgraded Y2H, CoIP, luciferase (LUC) complementation imaging assays. We then confirmed that both OsGF14h^{SN9816} and OsGF14h^{WR04-6} can interact with OsHOX3 and OsVP1 *in vivo* using Y2H and transient expression assay (Fig. 3a and b). These interactions can also be verified *in vivo* based on a pull-down assay (Fig. 3c). The results of co-immunoprecipitation and Y2H further proved that the interaction strength between OsGF14h^{WR04-6} and OsHOX3, and between OsGF14h^{WR04-6} and OsVP1 were greater than that of OsGF14h^{SN9816} *in vivo* (Fig. 3a and d). The supplemented ChIP-qPCR assay and upgraded EMSA further verified the binding capacity of OsHOX3 to the *OsPYL5* promoter (encoding ABA receptor) was enhanced by adding OsGF14h^{WR04-6} protein (Fig. 3f). Therefore, we hypothesized that OsGF14h^{WR04-6} inhibited the expression of *OsPYL5* through interaction with OsHOX3, thus inhibiting ABA signaling. We have upgraded these results in the revised MS.

We have replaced the subtitle of this paragraph with 'Haplotype distribution and evolution of *OsGF14h*'.

L227-228, the haplotype analysis is not informative to clarify the allele history, and I do not understand why the non-functional allele seems to be derived from Hap6. If the authors want to discuss this, simulation studies must be helpful to estimate allele age and differentiation timing of the cultivars. Maybe the authors can put Fig 5 and demonstrate the geographic distribution of alleles with the climate data.

Response: We are sorry for this confusion. Thanks so much for the suggestion. We first introduce a haplotype of *O.bathii* as an outgroup to re-construct the haplotype network and time-tree for detecting the allele age. The network showed that it only takes one step (the 4bp deletion) to convert Hap6 to Hap1. However, converting from other Haplotypes to Hap1 needs many more mutational steps (Fig. 5a). Time-tree shows the shortest genetic distance and closest divergence time estimated by Maximum Likelihood between hap1 and hap6 according to mcmctree method in PLAM software. Thus we cautiously speculate that ‘Hap1 was likely derived from Hap6 (*Or-IIIa* or *GJ-trp*)’.

Thank you for your good idea. As shown in Fig. 5, we drew a haplotype distribution map with average annual temperature, and the haplotype information is more clearly presented.

L231, do the authors mean "the Pi ratios suggested signatures of selective sweep on GF14h"? Please indicate the evidence to claim “there is selection.” Also, only the pi ratio is too weak to claim the selection. How about Tajima's D on Chr 11? The following article would help to look for selection signatures.

Ref. Li, M., Tian, S., Jin, L. et al. Genomic analyses identify distinct patterns of selection in domesticated pigs and Tibetan wild boars. Nat Genet 45, 1431–1438 (2013)

Response: Thanks for the good suggestions and reference article. we upgraded the window-based *Fst* analysis, added the window-based Tajima's *D* analysis, further optimized grouping, and reorganized the logic for this part. These population genomics parameters jointly demonstrate whether *OsGF14h* involves population differentiation, selection, and domestication. We itemize the main result as follows.

OsGF14h exhibited a significant differentiation signal between *GJ-tmp-weedy* and *GJ-tmp-cul* based on window-based *Fst* on chromosome 11 (Fig. 6a), which corresponds to the differentiation of haplotype frequency (Fig. 6a). However, both window-based Tajima' *D* (*GJ-tmp-weedy* and *GJ-tmp-cul*) and π ratios ($\pi_{Or-IIIa} / \pi_{GJ-tmp-weedy}$ and $\pi_{Or-IIIa} / \pi_{GJ-tmp-cul}$) do not support the selection and domestication occurred in *OsGF14h* region in both *GJ-tmp-weedy* and *GJ-tmp-cul* (Fig. 6a). The *OsGF14h* also exhibits population differentiation signals between *GJ-tmp-weedy* and *GJ-tmp-late*, and between *GJ-tmp-weedy* and *GJ-tmp-medium* based on window-based *Fst* analysis, and both of these differentiation signals overlapped with the selection signals of *OsGF14h* in *GJ-tmp-late* and *GJ-tmp-medium* defined by the window based Tajima' *D* respectively (Fig. 6b and 6c).

Interestingly, *OsGF14h* does not exhibit significant signals for the three parameters in *GJ-tmp-early* (Fig. 6d and Supplementary Fig. 10). It is worth noting that *GJ-tmp-early* mainly comprised of landraces, whereas *GJ-tmp-medium* and

GJ-tmp-late are modern cultivated varieties. Under this context, the *OsGF14h* exhibits a strong genetic divergence signal between weedy rice and modern cultivated rice but becomes weakening between weedy rice and landrace, implying that *OsGF14h*^{SN9816} plays a vital role in the genetic improvement of modern *japonica* rice, and the 4bp deletion could be the major causality.

L242-243, ‘The above evidence demonstrated that the selection and population differentiation were driven by the 4 bp deletion of *OsGF14h*.’ this conclusion is too much speculation. I do not see any evidence to lead this conclusion.

Response: Thanks for pointing it out. Based on multiple genomic parameters and in-depth interpretation in the revised MS, we conclude that *OsGF14h* is a promising candidate involved in the selection and population differentiation between cultivated rice and weedy rice in temperate *japonica* planting areas, and the 4bp deletion could be the major causality.

Regarding the connection between *OsGF14h* and GA synthesis, I think starting with *OsGA20ox1* is a kind of cherry-picking way. *Sdr4-k* and *Rc* are the same. How is RNA-seq for *gf14h* to look at genome-wide effects of *OsGF14h*? Since the authors have generated the transgenic lines, the experiments will not be difficult and time-consuming. Showing overviews of the gene function will strengthen the present study.

Response: Thanks for the valuable suggestions. Firstly, we created two new overexpression lines of WT^{SN9816} with its own CDS of *OsGF14h*^{SN9816} overexpressed (*OxOsGF14h*^{SN9816-5} and *OxOsGF14h*^{SN9816-11}), that showed no significant AG difference with that of WT^{SN9816} (Fig. 2d). The overexpression lines (*OxOsGF14h*^{WR04-6-8} and *OxOsGF14h*^{WR04-6-25}) showed significantly enhanced AG ability and seedling vigor compared with WT^{SN9816} (Fig. 2c, d, and e). We performed RNA-seq analysis for *OxOsGF14h*^{SN9816-5} and *OxOsGF14h*^{WR04-6-8} under both anaerobic and aerobic conditions to evaluate genome-wide effects of *OsGF14h*. We found that anaerobic conditions largely induced gene expression. Importantly, we found *OsGF14h*^{WR04-6} partially activated the gibberellin biosynthesis pathway and inhibited the ABA response pathway based on this RNA-seq. Then we cite the conclusions of related papers that “genes involved in gibberellin biosynthesis are crucial for rice development, especially *OsGA20ox1* plays a vital role at the initial growth stage¹⁴.”

On the other hand, based on the paper on *OsHOX3* cloning¹⁶ and new experimental evidence in the present study (Supplementary Fig.7), *OsHOX3* is a positive regulator of GA, and *OsGA20ox1* also exhibited the expected difference expression. We hope the above new results can strengthen the connection between *OsGF14h* and GA synthesis.

After careful consideration, we put the content of ‘*Sdr4-k* and *Rc*’ in the Discussion paragraph. Because recent studies demonstrate an additive effect of *Sdr4* and *Rc* on seed dormancy and Wang et al. (2022) provided conclusive biochemical evidence that *Rc* regulates preharvest sprouting (PHS) tolerance through

OsVP1. Therefore crosstalk between our new findings on OsGF14h with the above studies was discussed to infer their co-inheritance relationship⁷⁻⁹ (Line 365).

Minor comments

L221, Fig. 4a -> Fig. 5a

L231, Fig. 5b -> Fig. 5e?

Response: We are sorry for these mistakes and have corrected them in the revised MS.

About GWAS, which minor allele frequency did the authors use? Why were the first four PCs chosen for correction of the population structure? Were indels also used as the genetic markers?

Response: Thanks for pointing these out. We used a MAF value of 0.05 for genome-wide association analysis. We re-performed GWAS with the first two PCs because the best population structure was well dissected with the PC2 based on the PCA plot in the revised MS (Fig. 1d), and the GWAS result is significantly improved (Fig. 1b); thanks so much for pointing out this mistake. Yes, indels are also used as the genetic markers for GWAS. We updated the above data in the revised MS.

For Fig. 1-a, b, the Manhattan plots' labels do not correspond to the correct positions.

Response: We are sorry for this mistake and have corrected it in the revised MS.

For Fig. 1-b & d, does $-\lg(\text{p-value})$ mean $-\log_{10}(\text{p-value})$?

Response: We are sorry for this mistake and have corrected it in the revised MS.

Please use BULP for Fig1-c and f to show the corrected allelic effects.

Response: Thanks for the suggestion. The BLUP value calculated using sommer package in R shows the corrected allelic effects as shown in Fig.1c and h.

L95, how did the authors define haplotype? Please write down the methodology.

Response: Thanks for pointing this out. The haplotype-based association analysis of candidate genes (annotated by MSU Rice Genome Annotation Project Release 7) was conducted using CandiHap software by providing gff file downloaded from <http://rice.uga.edu/downloads.shtml>.

L98-102, what is the evidence that the authors concluded: "Os11g39540 is the strongest response to anaerobic treatment"? The methods and the explanation are insufficient. For Supplementary Fig. 1-g, if the y-axis shows the relative values to the control samples, 10% of the difference is not reliable due to the technical limitation of qRT-PCR. Also, the "relative expression level" of Os11g39370 is almost the same level with Os11g39540. I think this is not a big issue because the authors showed the gene effect by experiments in the following section, but clear explanations are still needed.

Response: Thanks for pointing this out. As shown in Supplementary Fig.2, *LOC_Os11g39370* is mainly expressed in leaves, whereas *LOC_Os11g39540*

(OsGF14h) is mainly expressed in seed based on two Spatio-temporal expression databases, <https://rapdb.dna.affrc.go.jp/index.html> and <http://bar.utoronto.ca/>. We hope this evidence can further help to nominate candidate genes.

L112, does OsGF14h of SN9816 bring complete loss-of-function? For natural variation, frameshift and stop codon after the functional domain do not always loss-of-function. I am curious the 4bp deletion truly brings a loss of functionality?

Response: Thanks for pointing out this critical issue. We fully agree with you that frameshift and stop codon after the functional domain do not always loss-of-function. We have supplemented and upgraded a number of experiments and analyses to demonstrate this issue (Fig.3 and Line 169-216). We hope that our new efforts and responses adequately address your concerns. Based on the prediction by AlphaFold2, the functional-type OsGF14h^{WR04-6} detected from weedy rice WR04-6 encodes a fully functional 14-3-3 protein that with 9 alpha-helix structure could form a clamp to bind the N-terminal loop of OsHOX3. The binding is mainly through the hydrophobic interactions between C-terminus of OsGF14h^{WR04-6} and N-terminus of OsHOX3. The OsGF14h^{SN9816} was detected mainly from temperate *japonica* cultivated rice, the frameshift mutation caused by the 4bp deletion in the coding region led to loss one helix in the C-terminus that weakened the strength of its interaction with OsHOX3, and two non-synonymous substitution SNPs that appear before the 4bp deletion were not located at the binding site. The other two synonymous substitution SNPs and one synonymous substitution SNP that appears after the stop codon may not affect the function. We then supplemented and upgraded several experiments to demonstrate how this natural variation finally affects the anaerobic germination. On the other hand, we found that overexpression of SN9816's own CDS did not enhance its germination ability. Based on the above evidence, we believe that the structural integrity of 14-3-3 protein determines its functional integrity. Considering that OsGF14h^{SN9816} also can interact with OsHOX3 and OsVP1, thus we defined *OsGF14h*^{SN9816} as a partial loss-of-function type in the revised MS.

Reviewer #4 (Remarks to the Author):

In the present manuscript, authors made an effort to identify novel regulators of waterlogging tolerance related to anaerobic germination (AG), particularly important in direct seeding rice cultivation practice. As a result, LOC_Os11g329540 associated to gene OsGF14h, a 14-3-3 protein has been identified. Authors claim that this protein acts as a hub influencing ABA and GA signaling, involved in rice stem elongation in response to submergence and in the development of other morphological adaptation to waterlogging. To this respect, authors report that OsGF14h represses ABA signaling through its interaction with OsHOX2 and OsVP1 which repress the ABA response, potentially at the level of the ABA receptor OsPYL5. OsGF14h overexpressors showed enhanced coleoptile growth and survival rate under anaerobic conditions, but no effect on germination in the presence of ABA has been shown. Conversely, knockouts of OsGF14h showed enhanced sensitivity to ABA in a germination assay in the presence of the hormone. In addition, OsGF14h seems to co-express with OsHOX3 and also physically interact as a result of a Y2H analysis. To this respect, OsHOX3 acts as a OsPYL5 transcriptional repressor binding to its promoter (as a result of a gel shift assay) also under normal conditions. Under anaerobic conditions expression of ABA-dependent genes is associated to the presence of an active OsGF14h. On the contrary, OsVP1 overexpressors exhibited an opposite behaviour to that of OsHOX3 overexpressors potentially indicating that these two genes might have contrasting roles, despite the actual role of OsVP1 has not been further investigated. Taken together, this part seems pretty convincing and the role of OsGF14h as a switch balancing anaerobic growth through manipulation of ABA and GA signaling is plausible. However, the actual role of OsVP1 is less studied and its involvement in the regulation of anaerobic germination in relation to OsGF14h and in balance with OsHOX3 is partially addressed.

Response: Thanks so much for the valuable comments that greatly improved our study and are very useful for improving the scientific logic and readability of this MS.

We did not provide one critical information in the initial manuscript that 14-3-3 protein can mediate the trans-regulation of the ABA-responsive factor OsEM by OsVP1, and it was added in the revised manuscript. On the other hand, intensive research has proven that OsVP1, as a vital TF, interacts with a bZIP factor, TRAB1, to inhibit germination by positively regulating the response sensitivity of rice to ABA. We supplemented and upgraded Y2H, CoIP, Pulldown, and luciferase (LUC) complementation imaging assays. We then confirmed that both OsGF14h^{SN9816} and OsGF14h^{WR04-6} can interact with OsVP1 *in vivo* and *ex vivo* based on Y2H, luciferase (LUC), and Pull-down (Fig. 3a, b, and c). The co-immunoprecipitation (CoIP) and Y2H further proved that the interaction strength between OsGF14h^{WR04-6} and OsVP1 was greater than that of OsGF14h^{SN9816} *in vivo* (Fig. 3a and d). *TRAB1-OsVP1* unit and other downstream ABA-responsive genes were suppressed significantly by OsGF14h under anaerobic conditions based on the new qRT-PCR results (Fig. 3g and Fig. 4a). Combining these new results with the solid genetic evidence from knockout lines of *OsVP1* on the regulation of anaerobic germination (Fig. 3f), we may infer that OsGF14h^{WR04-6} inactivates OsVP1 by the interaction.

1.- Moreover, ABA levels along with GA levels, need to be provided in OsGF14h OE and KO seeds and seedlings.

Response: Thanks for the valuable suggestions. A new overexpression line of WT^{SN9816} with its own CDS overexpressed, OxOsGF14h^{SN9816-5}, was also used to compare of endogenous ABA content. The ABA level of germinated seeds between OxOsGF14h^{SN9816-5} and OxOsGF14h^{WR04-6-8} (introduced the CDS of OsGF14h from WR04-6) showed no significant difference under the anaerobic stress. The knockout line *gf14h-15* also showed no significant difference in ABA content with the WT^{WR04-6} under anaerobic stress (Supplementary Fig.4). Cross talk with results of endogenous ABA assay, we believe OsGF14h acts as a signal switch to reducing ABA sensitivity.

At the seeding stage, As shown in the Figure below, we successfully detected bioactive gibberellin GA1 in seedlings among wild type, overexpression lines, and knockout lines under flooded conditions, however, no significant difference occurred. We believe that is due to ‘survivorship bias’ since the anaerobic stress response mainly occurs in the germination stage. Regardless of the genotype, once the seedlings are successfully formed, the growth and development tend to be the same (seeds either normal development or suffocated in the soil, with almost no intermediate form). Of course, the premise is that the rice tested in this study are all non-*Sub1A* genotypes (*Sub1A* is a favored gene in submerged-tolerant *indica*, whereas the chromosomal segment containing *Sub1A* is absent in the *japonica* subspecies). Therefore, to avoid the misleading of this survivorship bias, we will not present the results of GA and ABA at the seedling stage in our case if it is not mandatory.

Figure 1. Comparison of active gibberellin GA1 content

2.- Authors did not measure bioactive gibberellin levels: GA1, GA3, GA4 or GA7 and that is necessary to link with the physiological/morphological response. Despite it makes sense that *gf14h* plants show reduced GA20 albeit enhanced GA53 levels, bioactive GA levels are required to assess their impact on the physiological response.

Response: Yes, we agree with you that bioactive gibberellin levels are necessary to link with the physiological/morphological response. After many attempts, we got the unusual result that only GA3 was stably detected among four bioactive gibberellins based on the Multiple Reaction Monitoring method with the isotopel internal standard, although the difference of GA3 content among knockout lines, overexpression lines,

and the wild type were as expected. To avoid GA1 may have been mistaken for GA3 in our tested samples, we checked the substance information of GA1 and GA3 due to the structures between them are very similar, and the only difference between them is the single (GA1) or double (GA3) bond between C1 and C2. The two substances are separated by liquid chromatography, with the retention time of GA1 being 3.09 and GA3 being 2.96. Then the two substances are distinguished by ion pairs in mass spectrometry with the ion pair of GA1 being 448.3→389.2/241.1, and GA3 being 446.3→387.2/265.1, which means the two substances can be distinguished without confusion. However, rice cannot synthesize GA3 to our knowledge (although GA3 synthesis by rice was reported in rare cases). The active GAs that rice can synthesize are GA1 and GA4, whereas GA1 is synthesized during the vegetative growth stage, and GA4 is synthesized during the reproductive stage, especially in anthers. GA3 is a compound that rice cannot degrade by GA2oxidase, and it is physiologically impossible for rice to synthesize this compound as active GA, because GA signaling is permanently turned on by GA3. GA3 is synthesized by *Gibberella fujikuroi*, which infects rice and causes rice stupid seedling disease (unlimited growth and very tall seedlings).

The possible reason bioactive gibberellins failed to be stably detected is that antagonism of ABA and GA makes the bioactive GA not easily captured, especially ABA is dominant during germination. To avoid the controversy, we thought a safe option is not to include these results in the MS without sufficient evidence. We are sorry for this uncertainty.

Figure 2. Example of bioactive gibberellin determination

3.- Germination assays of *OsGF14h* OE in the presence of ABA are missing.

Response: Thanks for the valuable comments. We compared the relative germination rate (germination rate under the treatment of exogenous ABA divided by that of under standard conditions) for two kinds of OE lines, knockout lines, and their wild types. The results verified that ABA sensitivity of overexpression lines that introduced CDS of *OsGF14h*^{WR04-6} (*OxOsGF14h*^{WR04-6-8} and *OxOsGF14h*^{WR04-6-25}) were significantly

decreased compared with WT^{SN9816}, whereas the other two overexpression lines that introduced CDS of *OsGF14h*^{SN9816} (*OxOsGF14h*^{SN9816}-5 and *OxOsGF14h*^{SN9816}-11) also showed a consistent level of ABA sensitivity with the WT^{SN9816} (Fig. 4b). Thus, the conclusion that '*OsGF14h*^{WR04-6} reducing ABA sensitivity' is further strengthened. Thank you for this suggestion.

The rest of the work trying to link functional *OsGF14h* alleles with different germplasm accessions is less clear and needs some clarification. First, this part seems unconnected to the rest of the work, which is primarily focused on the characterization of the role of *OsGF14h* in the anaerobic germination of rice. I understand that this is part of a bigger project aimed at identifying QTLs related to enhanced aerobic germination in rice and interesting germplasm resources to introgress this trait into elite varieties. And that, of course, requires other studies to investigate the co-inheritance of other important quality traits (e.g. *Sdr4-n* and *Rc*).

Response: Thanks for the valuable comments. We will explain each of them point-by-point as follow.

1.- The mode of inheritance of functional *OsGF14h*, absent in SN9816 and Hap1 ecotypes, is rather speculative and requires the survey of more samples as the explanation on how functional *OsGF14h* found in Hap2 has rendered a non-functional allele in Hap1 is too speculative. To This respect, there is only one haplotype harboring a non-functional *OsGF14h* allele (Hap1), what happened to Hap9, also with a non-functional allele? This could clarify the mode of inheritance of that non-functional allele.

Response: Thanks for the valuable comments. Your concerns are very reasonable. We hope that our new efforts and responses adequately address your concerns. Based on the prediction by AlphaFold2, the functional-type *OsGF14h*^{WR04-6} detected from weedy rice WR04-6 encodes a fully functional 14-3-3 protein that with 9 alpha-helix structure could form a clamp to bind the N-terminal loop of *OsHOX3*. The binding is mainly through the hydrophobic interactions between C-terminus of *OsGF14h*^{WR04-6} and N-terminus of *OsHOX3*. The *OsGF14h*^{SN9816} was detected mainly from temperate *japonica* cultivated rice, the frameshift mutation caused by the 4bp deletion in the coding region led to loss one helix in the C-terminus that weakened the strength of its interaction with *OsHOX3*, and two non-synonymous substitution SNPs that appear before the 4bp deletion were not located at the binding site. The other two synonymous substitution SNPs and one synonymous substitution SNP that appears after the stop codon may not affect the function. We then supplemented and upgraded several experiments to demonstrate how this natural variation finally affects the anaerobic germination. Another non-functional type *GF14h* (Hap9), is a rare allele detected from *indica* subspecies, the frameshift mutation caused by the 1bp insertion led to the loss four helix, which resulted in a complete loss of binding sites with *OsHOX3*. On the other hand, we found that overexpression of SN9816's own CDS did not enhance its germination ability. Based on the above evidence, we believe that the structural integrity of 14-3-3 protein determines its functional integrity.

Considering that *OsGF14h*^{SN9816} also can interact with *OsHOX3* and *OsVP1*, thus we defined *OsGF14h*^{SN9816} as a partial loss-of-function type in the revised MS.

To further clarify the mode of inheritance of all kinds of alleles, we introduce a haplotype of *O.bathii* as an outgroup to reconstruct the haplotype network and time tree for detecting the allele age of *OsGF14h*. The network showed that it only takes one step (the 4bp deletion) to convert Hap6 to Hap1, however, converting from any other Hap to Hap1 needs many more mutational steps. Hap6 and Hap1 showed the closest divergence time based on a time-tree for the ten haplotypes of *OsGF14h* according to *mcmctree* method in PAML software. Thus we cautiously speculate that 'Hap1 was likely directly derived from Hap6 (*Or-IIIa* or *GJ-trp*)'. Another non-functional Hap9 is a rare allele which only detected in *indica* of Southeast Asia. The emergence of Hap9 maybe earlier than Hap1 based on the haplotype network and time tree. Therefore, we infer that the two non-functional haplotypes (Hap6 and Hap9) originate independently in the two rice subspecies. We also drew a haplotype distribution map with average annual temperature to make the haplotype information more clearly presented (Fig. 5c).

2.- The population differentiation index (*Fst*) is poorly explained. It is not clear what it means in this context: Is it that most SNPs are equally present with no clear bias and those close to 5 and 25 Mb are fixed in the population or come from one of the genotypes? What implications have the different threshold values in this population? Moreover, in the graphs, what is that gap between 20 and 25 Mbp? Is that the centromere? If that is the case that could have implications regarding the recombination of LOC_11g39540.

Response: Thanks so much for pointing out this critical issue. We are very sorry for this mistake. This 20Mb-23Mb genomic interval does not have a gap in our panel. The emergence of this gap is due to the server failure while calling the VCF files and accidentally skipping this 20Mb-23Mb genomic interval when rerunning. We have fixed this problem in the revised MS (Fig. 6 and Supplementary Fig. 8, 9 and 10).

To resolve the issue of '*Fst* is poorly explained', we upgraded the window-based *Fst* analysis, added the window-based Tajima's *D* analysis, further optimized grouping, and reorganized the logic for this part. Base on the upgraded reanalysis, the *OsGF14h* genomic region exhibits a strong genetic divergence signal between weedy rice and modern cultivated rice but becomes weakening between weedy rice and landrace, implying that *OsGF14h*^{SN9816} plays a vital role in the genetic improvement of modern *japonica* rice, and the 4bp deletion could be the major causality (Line 312-341, Fig. 6, and Supplementary Fig. 8, 9 and 10).

3.- In the final part of the manuscript, the co-inheritance of *Sdr4-n* and *Rc* genes, related to seed quality is interesting from the breeding point of view, but, in my humble opinion, out of the scope of the manuscript and poorly addressed as it stands. Moreover, the association of *Rc* with ABA signaling through *OsVP1* (which has also been poorly investigated in this work) requires more research to decipher the potential interaction with the rest of elements characterized, if any, and their interaction

mechanism. For instance, do *Rc* seeds have differential ABA sensitivity in germination assays than non-*Rc* seeds? This is probably part of another, bigger, piece of research.

Response: Thanks for the valuable comments. We also realized that, as research content, this part in the Results paragraph is out of the scope of this manuscript. However, as an essential gene involved in the selection and population differentiation, *OsGF14h* co-inheritance with other related genes is well worth discussing. After careful consideration, we decided to put this content in the Discussion.

Recent studies demonstrate an additive effect of *Sdr4* and *Rc* on seed dormancy, and Wang et al. (2020) provided conclusive biochemical evidence that *Rc* regulates preharvest sprouting (PHS) tolerance through *OsVPI*. These two studies are so timely and strongly support our point of view. Therefore crosstalk between our new findings on *OsGF14h* with the above studies was discussed to infer their co-inheritance relationship (Line 366-387).

Minor aspects

Figure 1a, chromosome position seems to be displaced, please amend.

L221 Fig. 5a not Fig. 4a.

Response: We are sorry for this mistake and have corrected it in the revised MS (Fig. 1b).

L255-256 the sentence is confusing, please rephrase. E.g.: [...] resulting in enhanced PHS,

Response: We are sorry for this confusion, and we have reorganized the entire paragraph (Line 366-387).

Reviewer #5 (Remarks to the Author):

Jian Sun and his colleagues developed a cultivated-weedy rice RIL population and map-based cloned a functional gene for flood adaptation. To my knowledge, this is the first case to find an important gene from weedy rice population, a special genetic resource for rice breeding. Their experiment of functional validation is solid and acceptable. I believe they will find more important genes from the weedy rice population in future.

Response: Thanks so much for the valuable comments that are very useful for improving readability of this MS.

Minor comments:

1. A paragraph in INTRODUCTION should be added to introduce weedy rice. I guess a lot of reviewers have no idea about it, for example its origin, distribution and population genomic studies recently, etc.

Response:

Thanks for the valuable comments. We have updated these information in the revised MS (Line 67-72).

2. In DISCUSSION section about weedy rice resource, they mentioned semi-domestication about weedy rice. I suggest to give more discussion on different views about origin of weedy rice based on the recent studies on weedy rice.

Response: Thanks for the valuable comments. We have given more discussion on different views on the origin of weedy rice based on the recent studies in the DISCUSSION section of the revised MS (Line 394-397).

References:

- 1 Zhang, X. *et al.* Rice pollen aperture formation is regulated by the interplay between OsINP1 and OsDAF1. *Nat Plants* **6**, 394-403, doi:10.1038/s41477-020-0630-6 (2020).
- 2 Zhai, K. *et al.* NLRs guard metabolism to coordinate pattern- and effector-triggered immunity. *Nature* **601**, 245-251, doi:10.1038/s41586-021-04219-2 (2022).
- 3 Yano, K. *et al.* Efficacy of microarray profiling data combined with QTL mapping for the identification of a QTL gene controlling the initial growth rate in rice. *Plant Cell Physiol* **53**, 729-739, doi:10.1093/pcp/pcs027 (2012).
- 4 Fukazawa, J. *et al.* Repression of shoot growth, a bZIP transcriptional activator, regulates cell elongation by controlling the level of gibberellins. *Plant Cell* **12**, 901-915, doi:10.1105/tpc.12.6.901 (2000).
- 5 Schoonheim, P. J., Costa Pereira, D. D. & De Boer, A. H. Dual role for 14-3-3 proteins and ABF transcription factors in gibberellic acid and abscisic acid signalling in barley (*Hordeum vulgare*) aleurone cells. *Plant Cell Environ* **32**, 439-447, doi:10.1111/j.1365-3040.2009.01932.x (2009).
- 6 Schultz, T. F., Medina, J., Hill, A. & Quatrano, R. S. 14-3-3 proteins are part of an abscisic acid-VIVIPAROUS1 (VP1) response complex in the Em promoter and interact with VP1 and EmBP1. *Plant Cell* **10**, 837-847, doi:10.1105/tpc.10.5.837 (1998).
- 7 Sugimoto, K. *et al.* Molecular cloning of Sdr4, a regulator involved in seed dormancy and domestication of rice. *Proc Natl Acad Sci U S A* **107**, 5792-5797, doi:10.1073/pnas.0911965107 (2010).
- 8 Wang, J. *et al.* Transcription Factors Rc and OsVP1 Coordinately Regulate Preharvest Sprouting Tolerance in Red Pericarp Rice. *J Agric Food Chem* **68**, 14748-14757, doi:10.1021/acs.jafc.0c04748 (2020).
- 9 Zhao, B. *et al.* Sdr4 dominates pre-harvest sprouting and facilitates adaptation to local climatic condition in Asia cultivated rice. *J Integr Plant Biol*, doi:10.1111/jipb.13266 (2022).

Reviewers' Comments:

Reviewer #1:

Remarks to the Author:

The authors have responded to all my questions well, and the manuscript has been improved significantly. The only query left is to please retain the original Figure 1a (right now is just a plain box). Thank you for all the revisions!

Reviewer #2:

Remarks to the Author:

I carefully read the revised manuscript and found that the authors have addressed most of my concerns. Some detailed results were still not explicitly explained.

1. The author should analyze the RNA-seq data to show that OsGF14h inhibit the ABA signaling.
2. Y2H and co-IP assays showed that the interaction of OsGF14hSN9806 and OsHOX3 or OsGF14hSN9806 and OsHOX3 is much weaker than that of OsGF14hWR04-6. However, this result is not confirmed by in vitro pull-down assay. Please comment on it.
3. Fig 2b. It should be better to change "Survival rate" to "Seeding/Germination rate".
4. P8L172. "ex vivo" to "in vitro".
5. In Fig4 OsGF14hWR04-6 "inhibits ABA sensitivity" should be "inhibits ABA response".

Reviewer #3:

Remarks to the Author:

The authors responded to my comments well and this manuscript has been largely improved, but some information is still missing in ms.

Please provide the numbers of RILs (n=168) in QTL mapping section. MAF threshold and genotype (genetic markers) information in GWAS, and haplotype definition are also missing.

Reviewer #4:

Remarks to the Author:

I think authors did an excellent job explaining all issues raised by this reviewer. There are a couple of aspects that need further clarification, though:

GA analysis

What kind of analysis did authors perform? it is mentioned that LC/MS following a MRM, but precursor-to-production transitions do not ring a bell to me.

GA1 [M-H]⁻ 347 and GA3 [M-H]⁻ is 345, being main product ions 229 and 143, respectively, as found in many methodological manuscripts. Other active GAs can also be analyzed (e.g. GA4 or GA7) in the same run.

I do not understand what is 448.3 in 448.3→389.2/241.1 and 446.3 in 446.3→387.2/265.1.

OsPYL5 expression (Fig 3k)

what happens to OsPYL5 expression in WTWR04-6? it is totally different from that of WTkitaake or WTSN9816. It has been shown that expression of certain subfamily II PYR/PYL/RCAR ABA receptors increases in response to soil flooding-induced hypoxic conditions in different plant species (Arbona et al. 2017 and De Ollas et al. 2021). Therefore, Fig. 3K-1 and Fig. 3K-3 are compatible with this, but not Fig. 3k-2. This needs further clarification.

The same for Fig. 4a, the different behaviour of lines needs further clarification.

ABA germination assay

WT rice seems quite insensitive to ABA, is that true? it is hard to believe that WT rice does not respond the slightest to ABA supply, whereas gf14h is hypersensitive, is it the same for hox3? conversely, OxGF14hWR04-6 should be insensitive. Have authors checked this?

Reviewer #5:

Remarks to the Author:

No further comments

REVIEWER COMMENTS

Reviewer #1 (Remarks to the Author):

The authors have responded to all my questions well, and the manuscript has been improved significantly. The only query left is to please retain the original Figure 1a (right now is just a plain box). Thank you for all the revisions!

Response: We are very glad and thankful that Reviewer are satisfied with the revisions to our manuscript. We are sorry that our original Figure 1a may not match your PDF reader due to formatting, we have use another image in the revised manuscript, I hope this issue can be fixed.

Reviewer #2 (Remarks to the Author):

I carefully read the revised manuscript and found that the authors have addressed most of my concerns. Some detailed results were still not explicitly explained.

Response: We are very glad and thankful that Reviewer are satisfied with the revisions to our manuscript. Thanks so much for the further valuable comments.

1. The author should analyze the RNA-seq data to show that OsGF14h inhibit the ABA signaling.

Response: Thanks for the suggestion. We reanalyzed the transcriptome data and compared the expression patterns of ABA signaling-related genes (GO:0009737) and ABA-responsive genes (GO:0009738). We found that OsGF14h inhibits ABA signaling more through its inhibition of ABA response by regulating ABA-responsive genes (GO:0009738), which also coincides with our experimental evidence. We have updated these results in the revised MS and Supplementary Fig. 6.

2. Y2H and co-IP assays showed that the interaction of OsGF14h^{SN9806} and OsHOX3 or OsGF14h^{SN9806} and OsHOX3 is much weaker than that of OsGF14h^{WR04-6}. However, this result is not confirmed by in vitro pull-down assay. Please comment on it.

Response: Thanks so much for your professional comment. We believe that both OsGF14h^{SN9816} and OsGF14h^{WR04-6} can interact directly with OsHOX3 and OsVP1 *in vitro* without difference in intensity, as confirmed by Pull-down assay. As a homologous of OsGF14h, GF14c bridges Hd3a and its interacting protein OsFD1 forming a crystal structure of a GF14c-OsFD1-Hd3a complex that regulates rice flowering (Taoka, et al., 2011). In the present study, we also speculate that the interaction of OsGF14h with the two proteins (OsHOX3 and OsVP1) *in vivo* may not be simple and direct interactions, but rather performs a function by forming ternary or multiplex complexes. Some biochemical reactions such as competitive binding may lead to differences in interaction strength between different forms of proteins, which deserves further in-depth study.

References:

Taoka, Ki., Ohki, I., Tsuji, H. et al. 14-3-3 proteins act as intracellular receptors for rice Hd3a florigen. *Nature* 476, 332–335 (2011). <https://doi.org/10.1038/nature10272>

3. Fig 2g. It should be better to change “Survival rate” to “Seeding/Germination rate”.

Response: We have changed “Survival rate” to “Seeding rate” in the revised MS.

4. P8L172. “ex vivo” to “in vitro”.

Response: We have changed “ex vivo” to “in vitro” in the revised MS.

5. In Fig4 OsGF14h^{WR04-6} “inhibits ABA sensitivity” should be “inhibits ABA response”.

Response: We have changed “inhibits ABA sensitivity” to “inhibits ABA response” in the revised MS.

Reviewer #3 (Remarks to the Author):

The authors responded to my comments well and this manuscript has been largely improved, but some information is still missing in ms.

Response: We are very glad and thankful that Reviewer are satisfied with the revisions to our manuscript. We are sorry that there are still some problems with our manuscript.

Please provide the numbers of RILs (n=168) in QTL mapping section. MAF threshold and genotype (genetic markers) information in GWAS, and haplotype definition are also missing.

Response: We are sorry for this missing information. The numbers of RILs (n=168), MAF threshold, and genotype information for GWAS were provided in appropriate parts in the revised manuscript (line 404, line 421, and line 451). The haplotype of each candidate gene was defined by all detected SNPs and indels within a region of 2,000 bp upstream to 500 bp downstream of annotated genes according to the default setting of CandiHap software (Line 460-Line 463).

Reviewer #4 (Remarks to the Author):

I think authors did an excellent job explaining all issues raised by this reviewer. There are a couple of aspects that need further clarification, though:

Response: We are very glad and thankful that Reviewer are satisfied with the revisions to our manuscript. We are sorry that there are still some problems with our manuscript.

1. GA analysis

What kind of analysis did authors perform? it is mentioned that LC/MS following a MRM, but precursor-to-product transitions do not ring a bell to me. GA1 [M-H]⁻ 347 and GA3 [M-H]⁻ is 345, being main product ions 229 and 143, respectively, as found in many methodological manuscripts. Other active GAs can also be analyzed (e.g. GA4 or GA7) in the same run. I do not understand what is 448.3 in 448.3→389.2/241.1 and 446.3 in 446.3→387.2/265.1.

Response:

Thanks so much for the professional comment. Yes, bioactive gibberellin levels are directly associated with the physiological and morphological response. Therefore they are also the primary targets that we want to measure. However, other bioactive gibberellins were not consistently detected in the present study (The reason, please refer to the R1 point-by-point response). As reviewer mentioned that other active GAs can also be analyzed (e.g. GA4 or GA7) in the same run, in fact our method can simultaneously detect 16 kinds of gibberellins, including four kinds of bioactive gibberellins.

There is no doubt that GA1 [M-H]⁻ 347 and GA3 [M-H]⁻ is 345, being main product ions 229 and 143, respectively, as found in many methodological manuscripts. The daughter and parent

ions of gibberellin mentioned in these papers are ion pairs of the gibberellin compound itself. In the present manuscript the ion pair mentioned are the ion pairs of the derivatization product formed by the derivatization of gibberellin (the example in figure below is from a paper published by Ting et al. 2017). After derivatization, the gibberellin derivatives carry charged groups that can increase the response to mass spectra.

Figure 2. Derivatization reaction of GA₁ with BPTAB reagent.

Fig. The example of derivatization reaction of GA1 showed by Ting et al. 2017

References:

Ting, Deng, Dapeng, Wu, Chunfeng, & Duan, et al. (2017). Spatial profiling of gibberellins in a single leaf based on microscale matrix solid-phase dispersion and precolumn derivatization coupled with ultraperformance liquid chromatography-tandem mass spectrometry. *Anal. Chem.*

2. OsPYL5 expression (Fig 3k)

What happens to OsPYL5 expression in WT^{WR04-6}? it is totally different from that of WT^{kitaake} or WT^{SN9816}. It has been shown that expression of certain subfamily II PYR/PYL/RCAR ABA receptors increases in response to soil flooding-induced hypoxic conditions in different plant species (Arbona et al. 2017 and De Ollas et al. 2021). Therefore, Fig. 3K-1 and Fig. 3K-3 are compatible with this, but not Fig. 3k-2. This needs further clarification. The same for Fig. 4a, the different behaviour of lines needs further clarification.

Response:

Thanks so much for your professional comment, and this is an interesting point. To answer this question, we conducted the new expression experiments. As shown in the figure below, most ABA-responsive genes exhibited the expected induction by flooding at 24h and 48h, which is consistent with your prediction and the results in the two papers (Arbona et al. 2017 and De Ollas et al. 2021). Based on our new results, we also believe ABA-responsive genes responding to hypoxia initiates in the early stages of germination. In terms of the physiological response to seed germination, the germination process is accompanied by an ABA-dominated to GA-dominated (or other development-related phytohormones) transition. In the present study, we detected ABA-responsive genes at the very beginning of germination (12h) under hypoxic conditions and detected the ideal significant difference between wild-type and transgenic lines/knockout lines. Although the signaling pathway of ABA was not sufficiently induced by flooding at the 12h (such as Fig.3k-2 and some cases in Fig.4a of the R1 manuscript mentioned by the reviewer), the *OsGF14h* has begun to perform a regulatory function, which is our priority concern.

Fig. Expression of ABA response-related genes at 24h and 48h

3. ABA germination assay WT rice seems quite insensitive to ABA, is that true? it is hard to believe that WT rice does not respond the slightest to ABA supply, whereas *gf14h* is hypersensitive, is it the same for *hox3*? conversely, *OxGF14h^{WR04-6}* should be insensitive. Have authors checked this?

Response:

The reviewer's concerns are very reasonable. The coleoptile of *WT^{WR04-6}* was indeed obviously inhibited by ABA instead of germination, and the germination rate showed the expected significant difference between wild-type and transgenic lines. Therefore we used the germination rate to reflect the ABA sensitivity.

Yes, we checked *hox3*, both coleoptile and germination are completely suppressed.

Yes, reviewer are correct. The ABA-sensitive level of *OxGF14h^{WR04-6}* was significantly reduced compared to its wild type, *WT^{SN9816}*. We are sorry that the vague designation between of the comparison groups in Fig4c may make reviewer mistake *WT^{WR04-6}* for the wild type, *WT^{SN9816}*. We have updated the Fig4c in the revised manuscript as shown below.

Fig. Comparison of relative germination rate (germination rate under the treatment of exogenous ABA divided by that of under normal conditions) among WT^{WR04-6}, knockout line (*gf14h-13*, *gf14h-15*), WT^{SN9816} and overexpression lines (OxOsGF14h^{SN9816}-5, OxOsGF14h^{SN9816}-11, OxOsGF14h^{WR04-6}-8, OxOsGF14h^{WR04-6}-25) under 72 hours of anaerobic incubation with exogenous 1 μM ABA treatment. Error bars represent mean ± S.D. (n = 3), according to Student's *t*-test.

Reviewer #5 (Remarks to the Author):

No further comments

Response: We are very glad and thankful that Reviewer are satisfied with the revisions to our manuscript.

Reviewers' Comments:

Reviewer #2:

Remarks to the Author:

The authors have addressed all my concerns. I have no further comments.

Reviewer #4:

Remarks to the Author:

Authors have satisfactorily addressed all my concerns. I have no further comments.